# EQUALLY CRITICAL: SAMPLES, TARGETS, AND THEIR MAPPINGS IN DATASETS

## ABSTRACT

Neural scaling laws highlight the trade-off between test error reduction and increased resources in machine learning, revealing diminishing returns as data volume, model size, and computational power increase. This inefficiency poses sustainability challenges, as marginal performance gains necessitate exponential resource consumption. Recent works have investigated these laws from a data-efficient standpoint, primarily concentrating on sample optimization, while largely neglecting the influence of target. In this study, we first demonstrate that, given an equivalent training budget, employing soft targets on a 10% subset can **outperform** the use of one-hot targets on the full dataset. Building on this observation, we review existing paradigms in the sample-target relationship, categorizing them into distinct sample-to-target mapping strategies. Subsequently, we propose a unified loss framework to assess their impact on training efficiency. Finally, we conduct a comprehensive analysis of *how variations in target and sample types, quantities, and qualities* influence training efficiency across three training strategies, providing *six key insights* to enhance training efficacy.

## 1 INTRODUCTION

Neural scaling laws (Hestness et al., 2017) manifest across various domains like vision (Hernandez et al., 2021; Zhai et al., 2022) and language (Kaplan et al., 2020; Hoffmann et al., 2022), indicating that test error often decreases following a power law relative to training data volume, model size, or computational resources. However, power law scaling is inherently inefficient. For example, in language modeling, reducing cross-entropy loss from approximately 3.4 to 2.8 nats demands tenfold more training data (Kaplan et al., 2020). Similarly, increasing pre-training data points from 1 billion to 3 billion yields only a modest accuracy gains on ImageNet (Zhai et al., 2022).

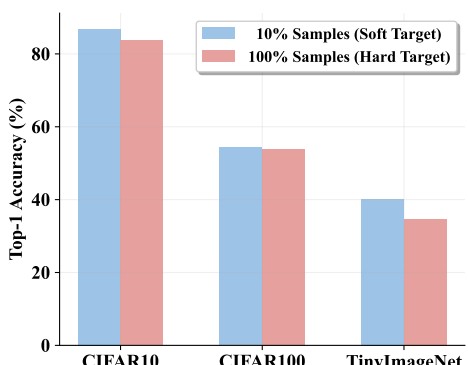

Figure 1: Under equivalent training budgets, performance across various datasets indicates that utilizing soft targets with only 10% of the original training samples can *surpass* the performance achieved using the complete dataset.

Recent studies (Sorscher et al., 2022; Fan et al., 2024) have explored neural scaling laws from a data-efficient learning perspective, showing that data pruning strategies (Paul et al., 2021) can outperform traditional power law scaling with reduced training budgets on pruned datasets. Additionally, studies have investigated scaling laws using synthetic (Fan et al., 2024) and augmented (Geiping et al., 2022) images.

In summary, *prior research predominantly focused on the sample component[1] of datasets.* However, advancements in data-efficient learning methods (Sun et al., 2024b; Yin et al., 2023; Guo et al.,

---

[1]In this paper, we decouple dataset into two essential components: samples $X$ and targets $Y$. For example, in a classification dataset, $X$ represents the images, while $Y$ consists of the one-hot encoded targets.

2024) have highlighted the importance of targets. For instance, efficient dataset distillation methods heavily rely on soft targets generated by pre-trained models (Sun et al., 2024b).Furthermore, Sun et al. (2024a) provide theoretical insights demonstrating that both targets and samples within a dataset significantly influence training dynamics and convergence.

Inspired by these findings, we conduct experiments to assess the effectiveness of soft targets, as depicted in Figure 1 . With the same training budget, models trained on a 10% subset with soft targets can **surpass** those trained on the full dataset with one-hot targets, emphasizing the potential of soft targets. This finding underscores the significance of the exploration the target component, a frequently overlooked aspect in the deep learning community.

In this paper, we conduct a comprehensive analysis of *how variations in target and sample types, quantities, and qualities affect training efficiency.* To enable a systematic examination of samples and targets, we first review existing paradigms in the sample-target relationship, categorizing them into distinct sample-to-target mapping strategies: multiple-to-one mappings (typical in conventional supervised learning), multiple-to-multiple mappings (common in knowledge distillation Hinton et al. (2015) using teacher models), and the proposed multiple-to-few mappings, as detailed in Section 3.1 . Each strategy exhibits unique training efficiencies, prompting a detailed investigation of their effects on training efficiency. we then propose a unified loss framework to evaluate their impact on training efficiency in Section 3.2 . Our analysis covers: (a) diverse target generation strategies in Section 4.2 and Section 4.3 , (b) various teacher model performance and data augmentation in Section 4.4 , and (c) sample quantity and quality in Section 5 . We list our **six key findings below**:

(a) Soft targets expedite early-stage training, but one-hot targets yield superior final accuracy.
(b) Stronger teacher models enhance final performance, but weak ones aid early learning.
(c) With sufficient training data, noise in soft targets can be reduced by our proposed STRATEGY C, improving final accuracy.
(d) Teacher models trained with augmentations like MixUp do not consistently improve student performance, while RandomResizedCrop adversely affects student training.
(e) With limited data, soft targets provide advantages, but larger sample sizes favor one-hot targets.
(f) RandomResizedCrop is most effective for one-hot targets, while Mixed-based augmentation excels with soft targets.

## 2 RELATED WORK

To assess the impact of data—consisting of samples $X$ and targets $Y$—on the learning, we revisit three distinct paradigms in deep learning and examine their respective data structures.

**Conventional representation learning.** Representation learning (Zhang et al., 2018a) enables models to automatically discover and extract meaningful features from raw data, facilitating tasks such as classification (Li et al., 2022), clustering (Ren et al., 2024), and segmentation (Minaee et al., 2021). These tasks typically rely on datasets with explicit sample-target pairs, such as ImageNet-1K (Deng et al., 2009). Recent studies (Radford et al., 2021; Hafner et al., 2021) have extended this framework by utilizing text-image pairs to train robust representation models. In these approaches, text is implicitly used as a target through contrastive learning (Chen et al., 2020b; Khosla et al., 2020). Moreover, self-supervised learning methods (Grill et al., 2020; Zbontar et al., 2021; Chen et al., 2021; Caron et al., 2021) have been developed to extract representations from unlabeled data by generating supervision signals through mechanisms like momentum models.

**Summary.** *Current representation learning techniques construct learning targets $Y$ from samples $X$, explicitly or implicitly, to guide models in learning the nonlinear mapping between them.*

**Soft label learning via knowledge distillation.** Knowledge distillation, introduced by Hinton et al. (2015), is a prominent technique for model compression in which a smaller student model learns from the outputs, or soft labels, of a larger teacher model. This approach effectively transfers knowledge with minimal performance degradation, enabling smaller, efficient models while maintaining capability (Gou et al., 2021). Subsequent research has expanded its application to enhance model robustness (Xu et al., 2020), facilitate domain adaptation (He et al., 2020b), and support semi-supervised learning (Chen et al., 2020a). Moreover, Yim et al. (2017) demonstrates that knowledge distillation can accelerate the optimization process during training.

**Summary.** *Knowledge distillation generates unique **soft targets** $Y$ for each sample via a pre-trained teacher model, embedding rich information that enables the student model to closely mimic the behavior of the teacher.*

**Data-efficient learning through condensed data.** Traditional dataset distillation aims to replicate the original dataset's behavior within a distilled dataset by minimizing discrepancies between surrogate neural network models trained on both synthetic and original data. Key metrics in this process include matching gradients (Zhao et al., 2020), features (Wang et al., 2022), distributions (Zhao & Bilen, 2023), and training trajectories (Cui et al., 2022). However, these methods incur substantial computational overhead due to the ongoing calculation of discrepancies, requiring numerous iterations for optimization and convergence. This makes them challenging to scale to large datasets, such as ImageNet (Deng et al., 2009). An effective strategy involves identifying metrics that capture critical dataset information (Yin et al., 2023; Sun et al., 2024b), thereby eliminating the need for exhaustive comparisons between original and distilled datasets. Consequently, these methods scale efficiently to large datasets, such as ImageNet-1K. For instance, SRe$^2$L (Yin et al., 2023) compacts the entire dataset into a model, such as pre-trained neural networks like ResNet-18 (He et al., 2016), and subsequently extracts knowledge into images and targets, forming a distilled dataset. Recently, RDED (Sun et al., 2024b) suggests that images accurately recognized by strong observers, such as humans and pre-trained models, are pivotal for effective learning.

**Summary.** *Dataset distillation methods extract key information from the original dataset, forming more informative samples $X$ and targets $Y$ that facilitate data-efficient learning.*

## 3 PRELIMINARY

This section begins by formally defining three mapping strategies between samples $X$ and targets $Y$. Subsequently, we critically analyze the limitations of existing loss functions in evaluating the impact of these mapping strategies on the representational capacity of models. We then introduce a novel unified loss function.

### 3.1 DEFINITION OF MAPPING STRATEGIES

Previous studies have demonstrated that data-efficient techniques, such as knowledge distillation and dataset distillation, which reconstruct the relationship between samples $X$ and targets $Y$, can significantly accelerate model training. For instance, knowledge distillation leverages soft targets generated by teacher models to expedite the training process (Yim et al., 2017). To thoroughly examine the relationship between samples $X$ and targets $Y$, we define mapping strategy, denoted as $\psi : \mathcal{X} \to \mathcal{Y}$, where $\mathcal{X}$ represents the data space and $\mathcal{Y}$ the target space. Importantly, augmentations are crucial in deep learning, significantly boosting training effectiveness (Zhang et al., 2018b; Geiping et al., 2022). Consequently, this paper emphasizes the use of augmented samples.

**STRATEGY A: Mapping multiple augmented samples within the same class into a one-hot target.** In conventional representation learning, data pairs $(X, Y)$ represent a non-injective mapping, denoted as STRATEGY A, as shown in Figure 2a. For example, for a classification task, one class $c$ may include multiple samples $X_c$, indicating that multiple samples are mapped to a same class. The mapping strategy is formalized as $\psi_{\mathrm{A}}(\mathbf{x}_i^{(j)}) = c$, where $j$ denotes the $j$-th augmented sample of the original sample $\mathbf{x}_i$.

**STRATEGY B: Mapping each augmented sample within the same class into a unique soft target.** For efficient knowledge distillation and dataset distillation, these approaches reconstruct the sample-to-target relationship using more informative soft targets. Each augmented sample corresponds to a unique soft target generated by a pre-trained model, establishing an injective mapping from augmented samples to soft labels, as shown in Figure 2b. Specifically, the mapping strategy is formalized as: $\psi_{\mathrm{B}}(\mathbf{x}_i^{(j)}) = \mathbf{y}_i^{(j)}$, where $\mathbf{y}_i^{(j)}$ represents the soft target generated by a pre-trained model for the augmented sample $\mathbf{x}_i^{(j)}$. It is obvious that STRATEGY B *introduces soft targets, which includes more information, thereby accelerating the training process of the model.*

**STRATEGY C: Mapping multiple augmented views of one sample into a same soft target.** While STRATEGY B provides valuable information through soft targets, it inevitably introduces noise by associating a single sample with multiple labels, potentially resulting in sub-optimal results. To

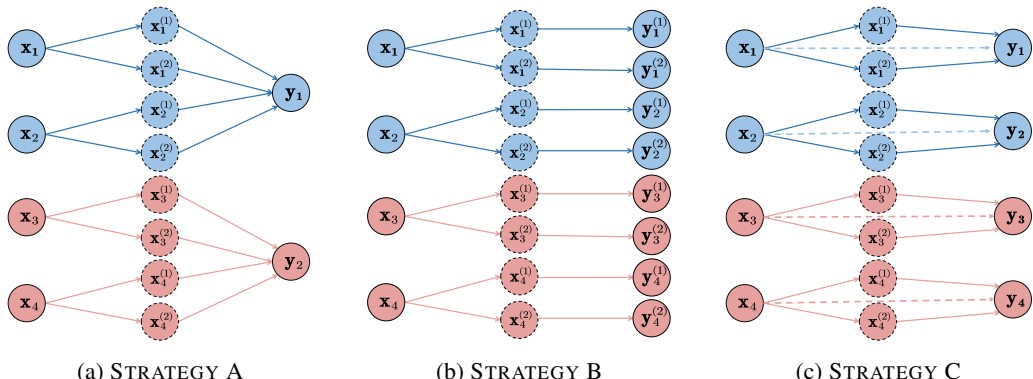

(a) STRATEGY A          (b) STRATEGY B          (c) STRATEGY C

Figure 2: **Three different sample-to-target mapping strategies.** STRATEGY A: Multiple augmented samples within the same class are mapped to one same one-hot target. STRATEGY B: Each augmented sample is mapped to a unique soft target. STRATEGY C: Multiple augmented views of a sample are mapped to one same soft target. Blue and red colors denote class 1 and class 2, respectively. For each strategy, $x_1$ and $x_2$ means original samples. The middle column illustrates the augmented samples produced by various augmentation strategies. On the left, $y_1$ and $y_2$ represent the targets associated with each augmented sample.

mitigate this issue while preserving the advantages of soft targets, we introduce a novel mapping strategy, STRATEGY C. This approach associates all augmented samples of an original sample with a single soft target, as shown in Figure 2c. It establishes a non-injective mapping from augmented samples to targets while maintaining an injective mapping from original samples to targets.

Specifically, the mapping function is defined as $\psi_C(x_i^{(j)}) = y_i$, where $y_i$ is generated by a pre-trained model for the original sample $x_i$. It preserves the rich information of soft targets while minimizing the noise introduced by multiple soft targets, which effectively balances the consistency of STRATEGY A with the variability of STRATEGY B.

### 3.2 DESIGN A UNIFIED LOSS FUNCTION

Having defined three distinct sample-to-target mapping strategies, we subsequently investigate their impact on the representational capacity of models. Following previous benchmarks and research (He et al., 2020a; Chen et al., 2020b; Grill et al., 2020), the representation ability of a trained model can be evaluated using an offline linear probing strategy, which involves decoupling the model into a backbone and a classifier. However, two inherent challenges emerge when employing conventional loss functions: (1) They inherently integrate these mapping strategies during training, complicating the evaluation of individual strategies. (2) They obscure the backbone's representational ability due to classifier influence.

Therefore, in this subsection, we first review conventional loss functions and analysis and their limitations. Subsequently, we propose a novel unified loss function to mitigate these challenges.

#### 3.2.1 CONVENTIONAL LOSS FUNCTIONS

Cross-entropy (CE) loss is widely recognized as a standard objective for training deep neural networks on classification tasks. However, its inability to exploit information from soft targets, which limits its application in mappings STRATEGY B and STRATEGY C. To use soft targets, Yang et al. (2022) introduces the Distillation Loss paradigm, which augments the CE loss with a soft-target matching component, thereby integrating soft label information into the model.

Despite this, the original CE loss continues to propagate one-hot target information to the model's backbone, implicitly introducing the multiple-to-one mapping to STRATEGY A. This characteristic complicates the independent analysis of each mapping strategy with respect to representation ability, which is unsuitable for our objectives. Furthermore, most knowledge distillation approaches (Gou et al., 2021; Xu et al., 2020; Zhao et al., 2022) treat the backbone and classifier as a unified entity during training. This coupling potentially obscures the backbone's feature extraction capabilities, as they may be inevitability affected by the classifier's performance. In summary, evaluating the impact of mapping strategies on representation ability of backbone remains challenging.

### 3.2.2 SEPARATION LOSS FUNCTION DESIGN

To address the challenges previously outlined, we propose a novel loss function designed to evaluate the influence of various strategies on representation ability. Specifically, we employ the Kullback-Leibler (KL) Divergence to leverage the information embedded in soft targets during the training of the backbone. For the classifier, we decouple it from the entire model and train it independently using cross-entropy (CE) loss. This approach prevents the transmission of one-hot target information from the classifier to the backbone, thus maintaining the separation between these mapping strategies. Furthermore, this decoupled training strategy ensures that the classifier performance does not adversely affect the representational capacity of backbone.

**Backbone Training.** For backbone training, denoted as $\phi_{\boldsymbol{\theta}}$, we employ the Kullback-Leibler (KL) divergence to align its predictions with the soft targets generated by the teacher models, thus effectively incorporating the STRATEGY B and STRATEGY C schemes.

To decouple the effect of the classifier, we introduce a novel softmax layer $\mathbf{g}$, projecting the outputs of $\phi_{\boldsymbol{\theta}}$ into soft targets. This layer replaces the classifier, thereby ensuring the classifier's isolation from the training process. The loss for backbone training is formulated as follows:

$$\mathcal{L}_{\mathrm{B}}(\phi_{\boldsymbol{\theta}}, \mathbf{g}) = \sum_{i=1}^{N} \sum_{j=1}^{n_i} D_{\mathrm{KL}} \left( \boldsymbol{\psi}(\mathbf{x}_i^{(j)}) \,\|\, \mathbf{g}(\phi_{\boldsymbol{\theta}}(\mathbf{x}_i^{(j)})) \right) , \tag{1}$$

where $N$ is the number of data points, $n_i$ is the number of augmented samples for $\mathbf{x}_i$, $\boldsymbol{\psi} \in \{\text{STRATEGY A}, \text{STRATEGY B}, \text{STRATEGY C}\}$ represents the mapping strategy, and $\mathbf{x}_i^{(j)}$ denotes the $j$-th augmented sample of the original sample $\mathbf{x}_i$.

**Classifier Training.** Following previous works (Gou et al., 2021; Xu et al., 2020), we employ the Cross-Entropy (CE) loss to leverage one-hot targets in training the classifier $\mathbf{h}$. The loss for classifier training is defined as follows:

$$\mathcal{L}_{\mathrm{H}}(\mathbf{h}) = -\sum_{i=1}^{N} \sum_{j=1}^{n_i} y_i \log(\mathbf{h}(\phi_{\boldsymbol{\theta}}(\mathbf{x}_i^{(j)}))) , \tag{2}$$

where $y_i$ is the one-hot target of the original sample $\mathbf{x}_i$.

Finally, the total loss function for model training is the sum of the backbone and classifier losses:

$$\mathcal{L}_{\mathrm{total}} = \mathcal{L}_{\mathrm{B}}(\phi_{\boldsymbol{\theta}}, \mathbf{g}) + \mathcal{L}_{\mathrm{H}}(\mathbf{h}) . \tag{3}$$

## 4 WHAT ROLE DO TARGETS PLAY? A COMPREHENSIVE INVESTIGATION OF MAPPING STRATEGIES

This section systematically investigates the impact of various mapping strategies $\boldsymbol{\psi} : \mathcal{X} \to \mathcal{Y}$ on the representational capacity of the backbone network $\phi_{\boldsymbol{\theta}}$. We begin by detailing the experimental setup in Section 4.1. Subsequently, we compare and analyze how different target types, namely one-hot targets in STRATEGY A and soft targets in STRATEGY B, affect the representational capacity of $\phi_{\boldsymbol{\theta}}$, as detailed in Section 4.2. Furthermore, under identical teacher model performance, we assess the effect of soft targets generated by distinct mapping strategies STRATEGY B and STRATEGY C, as discussed in Section 4.3. Finally, we explore the impact of soft targets produced by teacher models trained with diverse augmentation strategies, as described in Section 4.4.

### 4.1 EXPERIMENTAL SETUP

**Dataset and Network.** We conduct experiments using both large-scale and small-scale datasets, specifically CIFAR-10 (Krizhevsky et al., 2009), CIFAR-100 (Krizhevsky et al., 2009), and Tiny-ImageNet (Le & Yang, 2015). Our model utilizes ResNet-18 (He et al., 2016) as the backbone architecture, selected due to its widespread use and its generalizable performance across different architectures (Huang et al., 2017). The evaluation metric for all experiments is the Top-1 classification accuracy (%) on the test set. In this paper, we primarily present results on CIFAR-10, while results for CIFAR-100 and Tiny-ImageNet are provided in Appendix C.

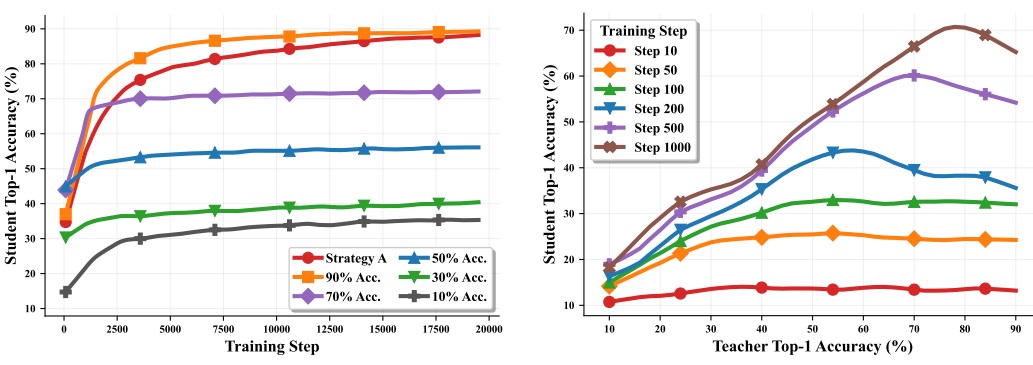

(a) Comparison STRATEGY A and STRATEGY B.  (b) Early-stage Training in STRATEGY B.

Figure 3: **STRATEGY B accelerates early-stage training but achieves inferior final performance. Left:** Comparison of STRATEGY A and STRATEGY B that employs teacher models of differing accuracies to evaluate the representational capacity of the model $\phi_\theta$. **Right:** Examination of the impact of training steps in the early training phase and the influence of teacher models with varying accuracies on STRATEGY B. All experiments are conducted on CIFAR-10 using the ResNet-18 architecture.

**Pre-trained Teacher Models.** To thoroughly investigate the influence of teacher models with varying accuracies on the representation capability of the student model $\phi_\theta$, we first train a series of teacher models with differing accuracies, employing various augmentations on the training set. Details of this process are provided in Appendix A.

Subsequently, we categorize the teacher models based on their performance: models achieving 90% or higher accuracy are classified as high-accuracy (strong), those with approximately 70% accuracy as medium-accuracy (moderate), and those with 50% or lower accuracy as low-accuracy (weak). These categories are representative enough to observe distinct patterns in the influence on $\phi_\theta$.

### 4.2 COMPARING ANALYSIS OF TARGET MAPPING: WHAT KINDS OF TARGETS DRIVES BETTER PERFORMANCE?

In this experiment, we primarily evaluate the influence of two distinct target generating strategies on performance of model $\phi_\theta$. Specifically, we consider STRATEGY A, which employs human-annotated one-hot targets, and STRATEGY B, which uses soft targets generated by pre-trained teacher models. Notably, STRATEGY C, similar to STRATEGY B and also using soft targets, will be discussed in Section 4.3. Moreover, for STRATEGY B, we also investigate how varying the quality of the teacher model affects model's performance.

**Comparison STRATEGY A and STRATEGY B.** The results, as presented in Figure 3a, demonstrate that *models trained with* STRATEGY B *exhibit accelerated learning during early training stages*. This is due to the additional information provided by soft targets, which enhances the training process. Furthermore, models leveraging higher-quality teacher models achieve greater accuracy. Nevertheless, models utilizing STRATEGY B ultimately exhibit inferior performance compared to those trained with standard augmentation (STRATEGY A), as indicated by the red line. This limitation arises because the guidance from teacher models imposes a ceiling on the student model's performance, preventing it from exceeding that of the teacher.

Conversely, models trained with the traditional STRATEGY A approach, despite slower improvements in the early-stage training, continue to improve performance over time and ultimately surpass the models trained with STRATEGY B.

**Claim 1.** *Utilizing* STRATEGY B *accelerates the early-stages training but ultimately limits accuracy in relation to teacher accuracy. Conversely, models trained with* STRATEGY A *exhibit slower initial convergence yet achieve superior final accuracy.*

**Early-Stage Learning: Stronger Teachers Are Not Always Optimal.** Previous experiments have demonstrated that the performance of student models is typically influenced by the quality of the

teacher model, with higher-performing teachers generally enhancing the final accuracy of the student model. However, it is crucial to note that *stronger teacher models are not universally advantageous.*

In this experiment, we focus on the early stages of training. We fix the number of training steps[2] to examine whether stronger teachers lead to greater improvements. As illustrated in Figure 3b, for a fixed number of training steps, as the teacher model's accuracy increases, there is an initial improvement followed by a decline in the performance of student models. This suggests that strong teacher models do not consistently provide significant advantages and may even hinder student model training during early-stage learning.

> **Claim 2 .** *While stronger teacher models generally enhance student model performance over the long term, weaker teacher models can facilitate early-stage learning, particularly when the number of training steps is limited.*

### 4.3 STRATEGY C: FEWER TARGETS, GREATER BENEFIT

In previous section, we primarily compared STRATEGY A and STRATEGY B. We now shift our focus to the comparison of STRATEGY B and STRATEGY C, both using soft labels generated by a pre-trained teacher model. The key distinction lies in the number of targets used during model training: STRATEGY B assigns different soft targets for each augmented sample, whereas STRATEGY C utilizes the same soft target for all augmented samples derived from the same original sample.

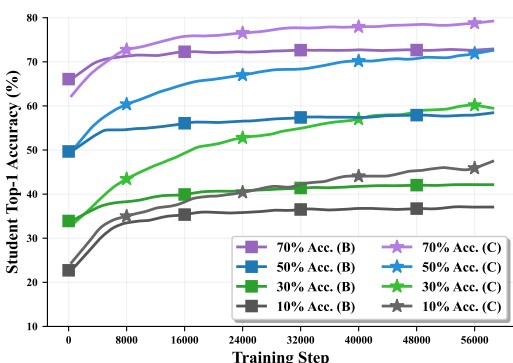

Figure 4: **Comparison of STRATEGY B and STRATEGY C.** STRATEGY C demonstrates superior performance in long-term training, surpassing STRATEGY B. Notably, even when using a randomly initialized teacher model with 10% accuracy, the student model trained with STRATEGY C outperforms the student model trained under STRATEGY B with a teacher model of 30% accuracy.

We conduct experiments to assess these strategies with teacher models of varying performance. As shown in Figure 4, the student model trained using STRATEGY C generally achieves higher accuracy. However, in the early training phase, STRATEGY B exhibits an advantage when the teacher model's accuracy is relatively high (e.g., 70%). As training progresses, the advantage shifts to STRATEGY C. We further conduct experiments employing a teacher model with different accuracy and compare the performance of the student model at different training stages under these two strategies, as illustrated in Figure 8 and Table 2 in Appendix B. We find that as the teacher model's accuracy increases from 10% to 90%, the improvement in student model performance initially rises and then declines. Notably, with a teacher model accuracy of 30%, the performance gain (a factor of 1.35 increase) achieved with STRATEGY C is substantially greater than that with a 90% accuracy teacher model (a factor of 0.99 increase). These results suggest that the benefit of STRATEGY C diminishes when the teacher model's accuracy is sufficiently high.

> **Claim 3 .** *Compared to STRATEGY B, STRATEGY C consistently achieves higher final accuracy with sufficient training steps. Notablty, during the early-stage training with a weak teacher model, STRATEGY C demonstrates a significant advantage.*

### 4.4 IMPACT OF DATA AUGMENTATION ON SOFT TARGET GENERATION

Prior sections have concentrated on the influence of teacher model performance on student model training. In this section, we focus on the intrinsic properties of teacher models. A notable study by Zhang et al. (2018b) highlights that advanced augmentation techniques like MixUp can enhance a model's generalization and robustness by creating new training samples through linear interpolation

---

[2]We also experimented with more training steps, see Table 1 in Appendix B for details.

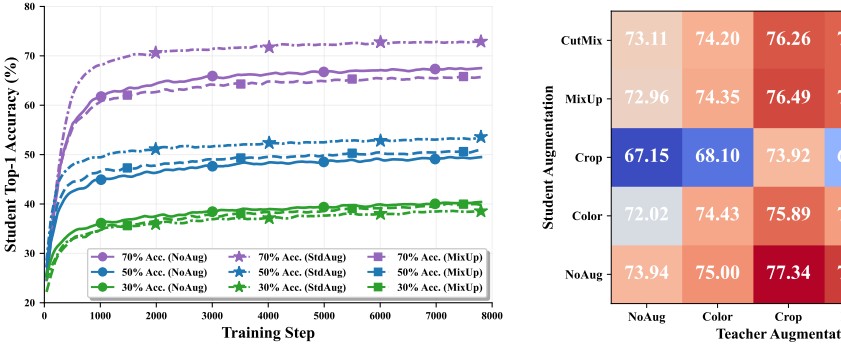

(a) Data Augmentations for Teacher Model   (b) Data Augmentations for Teacher and Student Model

Figure 5: **Left: Teacher models trained with various augmentation strategies.** Teacher models with low accuracy, trained using MixUp, can not help student model training. **Right: Applying various augmentations to both teacher and student models.** RandomResizedCrop significantly reduces student model performance.

between pairs of examples and their corresponding labels. Therefore, this section explores the influence of data augmentation techniques employed during teacher model training.

**Teacher Models with Various Data Augmentations.** In this experiment, we use teacher models with three identical performance levels (30%, 50%, and 70%) but trained with different augmentation strategies: (i) **NoAug:** no augmentation, (ii) **StdAug:** standard augmentation without MixUp, and (iii) **MixUp:** MixUp augmentation. Detailed descriptions of these strategies are provided in Appendix A. Additional augmentation strategies applied to the teacher model are discussed in Table 3 in Appendix B.

The results, as depicted in Figure 5a, demonstrate that *for high-accuracy teacher models, applying MixUp augmentation to the teacher does not significantly benefit the training of the student model.* Moreover, MixUp-trained teacher models performs the worst for a strong teacher. Interestingly, when teacher model accuracy is relatively low (e.g., 30% accuracy, as shown by the green line), the advantage of standard and MixUp strategies diminishes, and models trained without augmentation show comparable performance.

**Both student and Teacher Models with Various Data Augmentations.** To further assess the impact of augmentation techniques applied on student models themself, we fix the teacher model's accuracy[3] at 70% and train the student model using various data augmentations. The data augmentation techniques include five strategies: NoAug, RandomResizedCrop[4], StdAug, MixUp, and CutMix (Yun et al., 2019). Detailed descriptions of these augmentation strategies are provided in Appendix A.

The results, shown in Figure 5b, reveal that:

(a) For the teacher model, employing RandomResizedCrop, MixUp, or CutMix significantly enhances student model accuracy compared to other methods.

(b) For the student model, using RandomResizedCrop significantly reduces performance unless the teacher model was also trained with RandomResizedCrop.

> **Claim 4.** *(1) Teacher models trained with augmentation strategies such as MixUp can not consistently enhance student model performance, and when these teacher models have low accuracy, their ability to effectively support student models diminishes. (2) Additionally, employing RandomResizedCrop for student model training significantly impairs performance unless consistently applied to both teacher and student model training.*

## 5   How Does Sample Variation Affect Performance?

In preceding sections, we focus on how variations in target mappings impact the model training process. Notably, equally crucial to the performance of deep learning models is the quantity and

---

[3]Results for teacher models with different accuracy levels are available in Figure 9 in Appendix B.

[4]RandomResizedCrop is notably regarded as a strong augmentation method, alongside mix-based strategies.

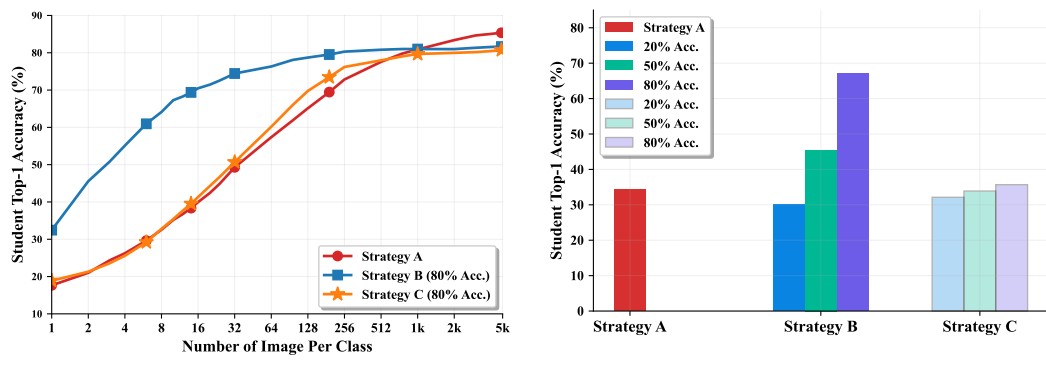

(a) Influence of `IPC`                    (b) Strategies Comparison with extremely small `IPC`

Figure 6: **Left: Student model accuracy scaling with increasing Images Per Class (`IPC`).** With limited IPC, STRATEGY B deviates from the typical power-law scaling compared with other strategies, whereas STRATEGY A exhibits notable advantages when `IPC` is sufficient. **Right: Strategy comparison when `IPC = 10`.** STRATEGY B demonstrates significant benefits, while STRATEGY C shows limited advantages.

quality of training samples. In this section, we first investigate how sample quantity affects model performance, followed by an analysis of sample quality. [5].

### 5.1 LESS IS MORE? PROBING DATA QUANTITY THROUGH IPC VARIATIONS

**Influence of `IPC` on Three Mapping Strategies.**    We begin by investigating how sample quantity influences model performance across three strategies using Images Per Class (`IPC`) as a metric. For each `IPC` configuration, student models are trained using these three strategies for 10,000 steps, a point at which models nearly achieve convergence. The results are depicted in Figure 6a. Key observations include:

(a) Both STRATEGY A and STRATEGY C exhibit typical *power-law scaling*. As the number of images per class increases, the rate of improvement significantly diminishes. Notably, STRATEGY C demonstrates a slight advantage over STRATEGY A when the sample size is limited, such as when `IPC` is below 500.

(b) For STRATEGY B, *limited `IPC` results in substantial performance gains*, achieving approximately 30% higher margins than STRATEGY A and STRATEGY C. However, as `IPC` increases, the benefits diminish. For example, when `IPC` surpasses 1K, the student model's accuracy approaches that of the teacher model and is eventually surpassed by STRATEGY A.

**Extremely small `IPC` Across Three Strategies.**    We further conduct experiments with the three strategies under extremely limited samples (e.g., `IPC = 10`). For STRATEGY B and STRATEGY C, we employ teacher models with three different performance levels (20%, 50%, and 80%), all trained with identical augmentation. As illustrated in Figure 6b, the results indicate that (1) with STRATEGY B, the student model nearly achieves the teacher model's accuracy, and teacher models with higher accuracy provide a greater advantage. (2) Conversely, with STRATEGY C, the student model performance exhibits negligible differences despite the varying accuracies of the teacher models.

> **Claim 5 .** *(1) In scenarios with low `IPC`, student models trained with* STRATEGY B *achieve significantly higher accuracy compared to* STRATEGY A *and* STRATEGY C. *Moreover, teacher models with higher accuracy provide a greater advantage in* STRATEGY B. *(2) Yet, as sample size increases, the traditional* STRATEGY A *demonstrates superior performance, as both* STRATEGY B *and* STRATEGY C *become limited by the teacher model's accuracy.*

### 5.2 STRATEGIC ENHANCEMENTS: DATA AUGMENTATION FOR SUPERIOR TRAINING

Beyond sample quantity, sample quality substantially impacts model performance. Data augmentation techniques (Perez & Wang, 2017; Shorten & Khoshgoftaar, 2019), such as random cropping, rotation,

---

[5]The accuracy of the teacher models in Section 5.1 and Section 5.2 is 80% and 90%, respectively. Additional experiments with different teacher model accuracies are detailed in Figure 10 and Figure 11 in Appendix B .

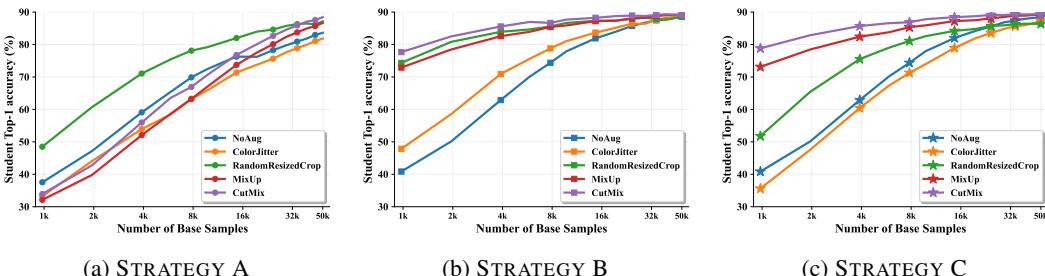

(a) STRATEGY A     (b) STRATEGY B     (c) STRATEGY C

Figure 7: **Scaling behavior of models using various data augmentation methods across three strategies.** For STRATEGY A, RandomResizedCrop enhances model performance significantly. In STRATEGY B and STRATEGY C, augmentation strategies surpass scaling law limitations when samples are limited. However, with sufficient samples, models converge to similar accuracy as the teacher model under both strategies, regardless of the augmentation used.

flipping, and color jittering, introduce controlled transformations that increase the diversity of training data. Geiping et al. (2022) investigates the relationship between data augmentations and scaling laws and quantify the value of augmentations. And in this section, we evaluates the impact of various data augmentation strategies[6] on model performance across three strategies. For each strategy, we vary the number of samples per class (IPC) and apply different data augmentation methods to assess their influence on model training. The results, depicted in Figure 7 , are as follows:

- STRATEGY A ( Figure 7a ): The influence of augmentation strategies varies with sample size. RandomResizedCrop significantly outperforms other augmentation strategies (ColorJitter, MixUp, CutMix) regardless of sample size. Notably, when the sample size is limited, no augmentation except RandomResizedCrop proves beneficial. However, when the sample size is sufficient, e.g., IPC exceeds 16K, CutMix and MixUp provide slight advantages over no augmentation.

- STRATEGY B ( Figure 7b ): All data augmentation strategies enhance model performance regardless of sample size. In particular, MixUp, CutMix, and RandomResizedCrop provide significant improvements with limited samples. Interestingly, with abundant samples, augmentation strategies converge to similar performance levels as that of the teacher model.

- STRATEGY C ( Figure 7c ): No Augmentation, ColorJitter, and RandomResizedCrop exhibit a performance pattern akin to STRATEGY A, while MixUp and CutMix show scaling behavior similar to STRATEGY B.

> **Claim 6 .** *(1) For* STRATEGY A*, RandomResizedCrop is the most effective augmentation method, particularly with limited sample sizes, whereas other strategies may not be beneficial. (2) Both* STRATEGY B *and* STRATEGY C *can surpass traditional* STRATEGY A *scaling law limitations through augmentation when data is scarce. With sufficient data, models utilizing* STRATEGY B *converge to similar performance levels regardless of the augmentation method employed.*

# 6  CONCLUSION AND FUTURE WORK

**Conclusion.**    To the best of our knowledge, this study is the first to emphasize the critical role of target in breaking neural scaling power laws. We present three sample-to-target mapping strategies and propose a unified framework to evaluate their effects on training processes. Our analysis investigates the impact of target and sample types, along with their quantities and qualities, on training efficiency. Based on this analysis, we provide six key insights for improving training efficiency.

**Future Work.**    In future work, we aim to perform a comprehensive analysis of these phenomena utilizing more complex and large-scale datasets, such as text data. By examining results across diverse datasets and employing advanced networks and algorithms, we will assess the robustness and generalizability of our findings.

---

[6]Detailed descriptions of these augmentation strategies are provided in Appendix A .

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

## A  EXPERIMENTAL SETUP

All experiments in the main body were conducted using the CIFAR-10 dataset. By default, the number of images per class in the training set is 5000, except for experiments in Section 5.1 , Section 5.2 . For all experiments throughout the paper, samples were randomly shuffled at the beginning of each epoch, and the batch size was fixed at 128. The number of batches per epoch after applying drop last strategy is 390 for both CIFAR10 and CIFAR100, and 780 for TinyImageNet. For the experiments in Figure 3 and 13, training was run for 50 epochs (corresponding to 20,000 steps for CIFAR10 and CIFAR100, and 40,000 steps for TinyImageNet), with validation averaged 10 times per epoch. For the experiments in Figure 5a and 16, training was run for 20 epochs (7,800 steps for CIFAR10 and CIFAR100, and 15,600 steps for TinyImageNet), also with validation averaged 10 times per epoch. For the experiments in Figure 4 and 8, training was run for 150 epochs (i.e., 58,500 steps), with validation performed once at the end of each epoch (i.e., every 390 steps). And for experiments in Section 5.1 , Section 5.2  and Figure 1, 5b, 9, 10, 11, 17, 18, the total number of training steps is set to fixed 10,000 steps.

For the experiments on CIFAR10, CIFAR100, and TinyImageNet, both the teacher and student models were implemented using a modified ResNet-18 architecture (no pretrained weights) where the initial convolutional layer was replaced with a $3 \times 3$ convolutional layer, using stride 1 and padding 1 to better accommodate the input image sizes. The batch normalization layer was adjusted accordingly to match the new convolutional output, and the max-pooling layer was removed. To better evaluate the backbone of the model, we pruned the final fully connected layers, and a linear layer was attached from the output feature dimension to the number of classes in the corresponding dataset, the backbone of the model and the attached classifier layer were trained separately, as described earlier in Section 3 .

For the teacher models used in the experiments, we employed the traditional STRATEGY A strategy (i.e., using one-hot targets) and trained them with the standard cross-entropy loss function. Validation was performed at the end of each step, and the teacher model was saved the first time it reached the preset accuracy, which was then used for subsequent student model training, and we ensure that the validation error for each teacher model with a specific accuracy is less than 1%. For STRATEGY B and STRATEGY C, the student model is trained using the soft targets by the teacher model, with the temperature of the soft labels set to 2.0.

All data points in the figures and tables of this paper were obtained by running each experiment at least five times and averaging the results. To reduce noise and provide clearer visualization of long-term trends, we applied locally weighted scatterplot smoothing (LOWESS) (Cleveland, 1979) to the figures that examine long-term behavior, such as Figures 3, 4, 8, 13, 14, and 16, using 10% of the data points for each local regression. Considering that the curve fitting applied is not well-suited for observing early-stage data, particularly when assessing the impact of STRATEGY C in the initial phases, we provide additional numerical results in the early training phase in Table 2, 5, 6, 7.

We employed the default implementation of the AdamW optimizer in the PyTorch Lightning Module, where the learning rate is set to $1 \times 10^{-3}$, and the weight decay is 0.01. The metric for evaluating the performance of all models is their Top-1 validation accuracy on the test set. Experiments were conducted on NVIDIA RTX 4090 GPUs and Intel Xeon processors.

We define the data augmentation techniques mentioned throughout the paper as follows:

**No Augmentation (NoAug):** Only normalization is applied based on the corresponding dataset.

**Standard Augmentation (StdAug):**

- **ColorJitter (or Color in short)**: Apply random changes to the brightness, contrast, saturation with a probability of 0.8 to each image.
- **RandomGrayscale**: Converts the image to grayscale with a 0.2 probability.
- **RandomHorizontalFlip**: Flips the image horizontally with a 50% chance.
- **RandomResizedCrop (or Crop in short)**: Resizes and crops the image to the input size with a scaling factor between 0.08 and 1.0.

**Mix-based Augmentation (not include StdAug in our settings):**

- **MixUp**: Perform a linear interpolation between two randomly selected images, where the interpolation factor, $\lambda$, is drawn from a Beta distribution with parameters $(\alpha, \beta) = (0.8, 0.8)$ which controls the degree of interpolation between the two images.

- **CutMix**: A bounding box is randomly generated within one image, and the pixel values inside the box are replaced by the corresponding region from a second randomly selected image. The area of the bounding box is determined by a parameter sampled from a Beta distribution with parameters $(\alpha, \beta) = (1.0, 1.0)$ which controls the relative size of the cutout region and the pasted-in region.

For all experiments except in Section 4.4, Section 5.2 and Tabel 3, 8, Figure 9, 16, 18, we employed the **Standard Augmentation (StdAug)** and normalization for both teacher models and student models. For the experiments in Figure 5a, 16, and Table 8, we applied **StdAug** to the student model. For the experiments in Figure 7 and 18, **StdAug** was applied to the teacher model.

## B  ADDITIONAL EXPERIMENTAL RESULTS

We further conducted more experiments on the CIFAR10 dataset to supplement and verify the patterns observed in the main body. We first show the experimental patterns can be observed more clearly under the step setting in Section 4.2 through Table 1. And we compare STRATEGY B and STRATEGY C using more teachers during different training stages to observe under what conditions can STRATEGY C hold a greater advantage through Table 2, and we find that STRATEGY C will not yield a significant advantage when the accuracy of the teacher model is high enough (see Figure 8).

In addition to the data augmentation strategies used in Section 4.4, we further compared the effects of applying more data augmentation strategies to teacher models with different accuracies in Table 3. We find that in most cases only when applying the RandomResizedCrop strategy to the teacher model does it significantly help the training of the student model, while most other individual augmentation strategies tend to have a negative impact.

Aside from the investigation of samples in Section 5, we further conducted more experiments and found that the scaling behavior of student models trained by teacher models with different accuracies shows slight differences, as shown in Figure 10 and 11.

In addition to the experiments above, we performed ablation studies on the effects of batch size and learning rate, as shown in Figure 12.

All values presented in the tables represent the Top-1 accuracy of the student model under the corresponding teacher model settings with the percentage symbol (%) omitted for simplicity.

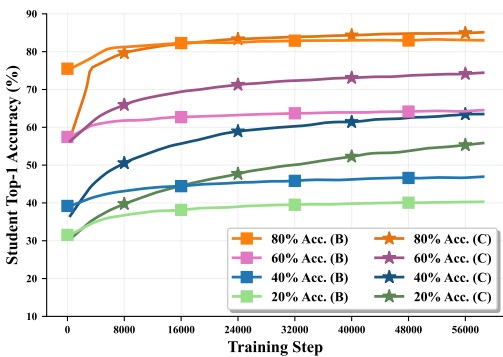

Figure 8: **Comparison of STRATEGY B and STRATEGY C under more different teacher models with different accuracies.** It further shows that as the accuracy of the teacher model increases, the advantage of using STRATEGY C over STRATEGY B becomes less apparent.

Table 1: **The Top-1 accuracy of the student model under different training steps and different teachers.** The **bolded** data represent the maximum student Top-1 accuracy in each training step. Only under a limited number of training steps (i.e., 1000 steps setting in the main body), the teacher model with relatively medium accuracy will show its advantage.

| Training Step | Teacher Top-1 Accuracy (%) $\pm1\%$ | | | | | | | | |
|---|---|---|---|---|---|---|---|---|---|
| | 10% | 20% | 30% | 40% | 50% | 60% | 70% | 80% | 90% |
| 10 | 10.59 | 12.92 | 13.39 | 14.08 | 13.86 | **15.53** | 14.25 | 15.14 | 14.38 |
| 20 | 11.85 | 18.57 | 18.79 | 18.47 | **19.51** | 19.02 | 19.11 | 17.94 | 19.37 |
| 50 | 13.46 | 20.94 | 23.54 | 24.97 | **27.69** | 26.64 | 27.46 | 26.42 | 25.52 |
| 100 | 14.76 | 23.19 | 26.71 | 31.15 | **35.24** | 32.55 | 32.32 | 33.00 | 32.22 |
| 150 | 15.63 | 25.59 | 27.95 | 33.38 | 37.77 | **37.81** | 36.57 | 36.52 | 34.66 |
| 200 | 18.52 | 26.23 | 31.11 | 36.10 | 43.29 | **43.31** | 41.32 | 39.44 | 37.93 |
| 300 | 19.16 | 27.66 | 32.00 | 36.47 | 46.73 | **51.51** | 50.69 | 46.67 | 42.99 |
| 500 | 19.49 | 29.26 | 33.43 | 38.06 | 48.93 | 55.72 | **61.06** | 57.68 | 54.79 |
| 700 | 21.99 | 29.84 | 33.59 | 38.73 | 48.83 | 56.50 | 63.71 | **65.16** | 62.27 |
| 1000 | 24.38 | 31.17 | 33.57 | 39.32 | 49.63 | 58.38 | 65.91 | **70.13** | 67.90 |
| 1200 | 24.50 | 32.35 | 34.98 | 40.64 | 51.56 | 58.81 | 67.45 | **73.09** | 70.23 |
| 1500 | 25.88 | 32.30 | 35.31 | 39.92 | 51.76 | 58.79 | 68.00 | **75.29** | 73.92 |

Table 2: **Comparing the student models trained with STRATEGY B and STRATEGY C at different training stages.** The Gain represents the performance improvement factor of the student model using STRATEGY C compared to STRATEGY B, with **bold** represent cases where the gain is greater than 1 (i.e., STRATEGY C is more advantageous). For a high-accuracy teacher model, STRATEGY C does not show advantages with fewer training steps. However, for a low-accuracy teacher model, training advantages are observed in both early and late training stages. And as the accuracy of the teacher model increases from 10% to 90%, the gain of using STRATEGY C compared to STRATEGY B first increases then gradually decreases.

| Step | Strategy | Teacher Top-1 Accuracy (%) $\pm1\%$ | | | | | | | | |
|---|---|---|---|---|---|---|---|---|---|---|
| | | 10% | 20% | 30% | 40% | 50% | 60% | 70% | 80% | 90% |
| 1950 | B | 25.09 | 33.31 | 35.93 | 40.29 | 51.63 | 59.12 | 66.93 | 73.31 | 72.80 |
| | C | 27.06 | 32.54 | 34.79 | 41.28 | 52.95 | 57.55 | 63.14 | 67.99 | 68.66 |
| | Gain | **1.08** | 0.98 | 0.97 | **1.02** | **1.03** | 0.97 | 0.94 | 0.93 | 0.94 |
| 7800 | B | 32.02 | 36.66 | 38.35 | 43.19 | 54.44 | 61.76 | 71.24 | 81.27 | 87.33 |
| | C | 34.07 | 39.86 | 42.96 | 50.57 | 60.68 | 65.89 | 72.47 | 79.67 | 84.00 |
| | Gain | **1.06** | **1.09** | **1.12** | **1.17** | **1.12** | **1.07** | **1.02** | 0.98 | 0.96 |
| 39000 | B | 35.96 | 39.70 | 41.61 | 46.11 | 57.18 | 63.90 | 72.58 | 82.93 | 90.35 |
| | C | 42.86 | 51.98 | 55.96 | 61.34 | 69.84 | 73.02 | 77.93 | 84.29 | 89.67 |
| | Gain | **1.19** | **1.31** | **1.35** | **1.33** | **1.22** | **1.14** | **1.07** | **1.02** | 0.99 |

## C  ADDITIONAL DATASET

We further conducted experiments on different datasets, including CIFAR100 and TinyImageNet to validate the findings presented in the main body. We first compared STRATEGY A and STRATEGY B on the two datasets in Figure 13 and 14, and further investigated the behavior of STRATEGY C through Table 5. We also compared the impact of applying different image augmentation strategies to the teacher models in Figure 16 and Table 8, and further explored the scaling behavior influenced by changing the sample size and applying different augmentation strategies on the student models, as shown in Figure 17 and 18. To have a more comprehensive evaluation on the differences between our proposed strategy STRATEGY C and traditional methods, we conducted experiments on the ImageNet

Table 3: **Comparison of different augmentation method applied for the teacher model using STRATEGY B (training step=7800).** The **bolded** data represent the maximum student Top-1 accuracy in each column. Besides the advantage of NoAug over MixUp in the teacher model shown in section 4.4, this table further indicates that applying RandomResizedCrop to the teacher model will benefit the training of the student model, regardless of the accuracy of the teacher model.

| Augmentation (on teacher) | Teacher Top-1 Accuracy (%) ±1% | | | | | | | |
|---|---|---|---|---|---|---|---|---|
| | 10% | 20% | 30% | 40% | 50% | 60% | 70% | 80% |
| NoAug | 31.20 | 35.55 | 40.29 | 43.92 | 49.50 | 56.94 | 67.59 | 77.43 |
| ColorJitter | **32.18** | 35.74 | 38.44 | 41.61 | 50.06 | 58.83 | 65.88 | 77.37 |
| Crop | 31.14 | **37.43** | **41.27** | 42.93 | **54.26** | **66.11** | **75.37** | **82.28** |
| MixUp | 30.30 | 35.00 | 39.81 | 43.79 | 50.93 | 61.44 | 65.79 | 75.78 |
| CutMix | 30.74 | 35.86 | 37.78 | 41.23 | 48.41 | 55.72 | 68.65 | 73.24 |
| StdAug | 31.53 | 35.65 | 38.49 | **49.52** | 53.57 | 63.34 | 73.00 | 80.59 |

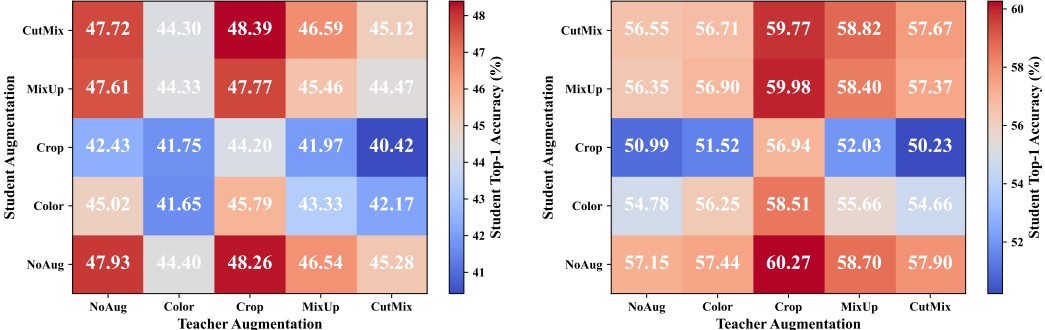

Figure 9: **Ablation experiments of training the student model with 30% (Left) and 50% (Right) teacher models in the main body.** It is evident that using the RandomResizedCrop strategy has certain disadvantages for the student model. Additionally, applying the MixUp strategy to teacher models with lower accuracy undermines the effectiveness of training the student model compared to NoAug.

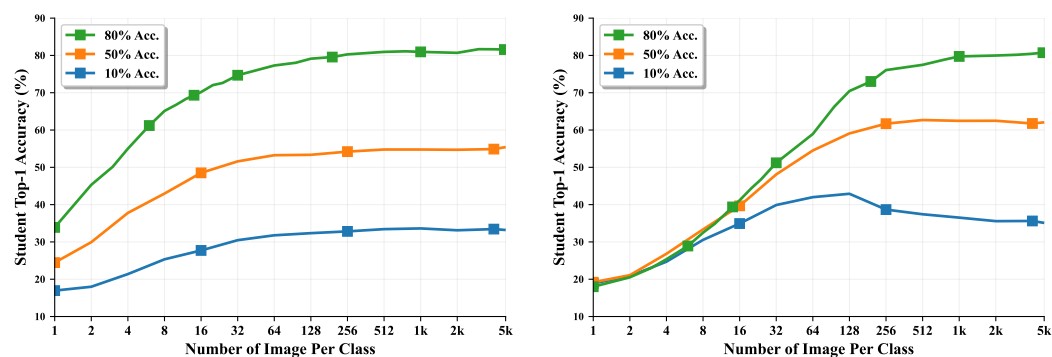

Figure 10: **The scaling behavior across different teachers (by changing the number of image per class) under STRATEGY B (left) and STRATEGY C (right).** In STRATEGY B, different teacher models exhibit similar patterns of scaling, whereas in STRATEGY C, the performance of the student model trained by a randomly initialized teacher model (i.e., 10% Acc.) initially improves but then declines as the IPC increases.

dataset, which is larger and more representative of real-world scenarios. We selected ResNet50 and ViT as the backbone architectures for teacher and student models, and batch size was set to 512, and we validated every 500 steps, with other hyperparameters remian unchanged. The visual results are shown in Figure 15, while detailed numerical results are provided in Table 6, 7.

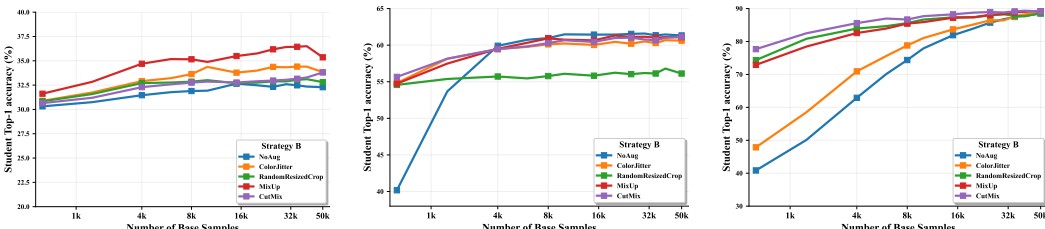

Figure 11: **The scaling behavior across different teacher using different data augmentation under STRAT-EGY B.** From left to right, the teacher models have accuracies of 10%, 50%, and 90%, respectively. For a teacher model with moderate or high accuracy, applying different data augmentation strategies to the student model results in their performance converging to nearly the same value (except under the 50% teacher model where using the RandomResizedCrop strategy for the student model is insensitive to changes in the number of samples). However, for a low-accuracy teacher model (in this case, a completely random initialized teacher model), applying different data augmentation strategies to the student model shows slight variations in effect as the number of samples increases.

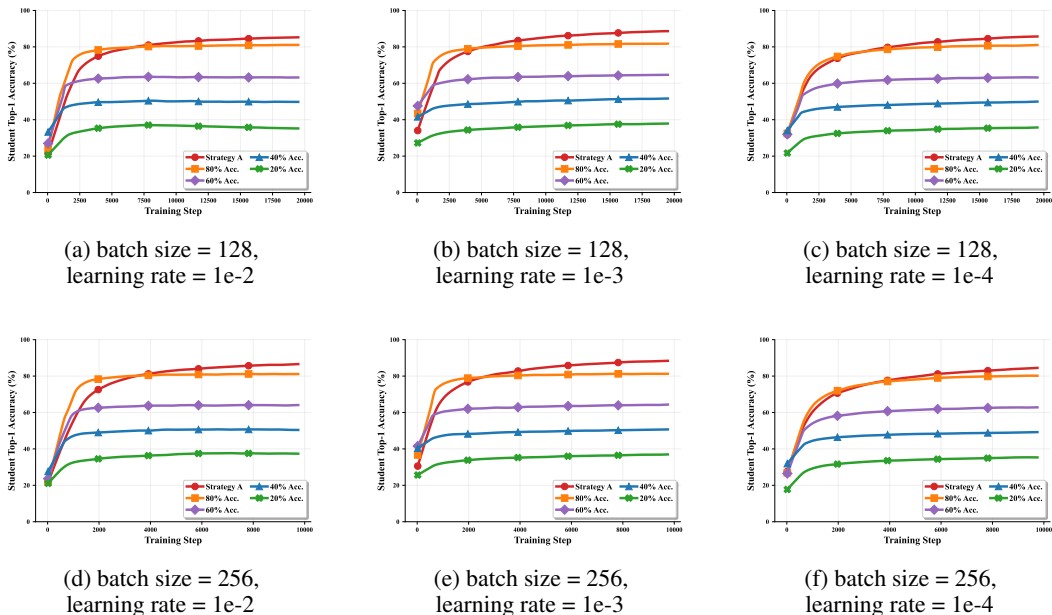

| (a) batch size = 128, learning rate = 1e-2 | (b) batch size = 128, learning rate = 1e-3 | (c) batch size = 128, learning rate = 1e-4 |
|---|---|---|
| (d) batch size = 256, learning rate = 1e-2 | (e) batch size = 256, learning rate = 1e-3 | (f) batch size = 256, learning rate = 1e-4 |

Figure 12: **The comparison of strategies STRATEGY A and STRATEGY B under different learning rate and batch size settings.** All results show that the STRATEGY B has a short term advantage while STRATEGY A have a long term advantage.

Table 4: **The results of student models trained with different batch sizes and learning rates under STRATEGY A and STRATEGY B, evaluated across varying levels of teacher model accuracy.** For each configuration, we report the mean Top-1 accuracy of the student model along with the standard deviation, calculated over five independent runs.

| batch size | learning rate | Teacher Top-1 Accuracy (%) $\pm 1\%$ | | | | |
|---|---|---|---|---|---|---|
| | | 20% | 40% | 60% | 80% | NoTeach |
| 128 | 1e-2 | $35.42 \pm 0.48$ | $49.54 \pm 0.93$ | $63.23 \pm 0.53$ | $81.18 \pm 0.39$ | $85.52 \pm 0.47$ |
| | 1e-3 | $37.86 \pm 0.32$ | $51.44 \pm 0.30$ | $64.54 \pm 0.79$ | $81.68 \pm 0.30$ | $88.59 \pm 0.48$ |
| | 1e-4 | $34.88 \pm 0.47$ | $49.63 \pm 0.46$ | $63.20 \pm 0.73$ | $81.09 \pm 0.33$ | $86.01 \pm 0.42$ |
| 256 | 1e-2 | $37.39 \pm 0.32$ | $50.70 \pm 0.37$ | $64.42 \pm 0.59$ | $81.78 \pm 0.40$ | $86.50 \pm 0.66$ |
| | 1e-3 | $37.06 \pm 0.56$ | $50.35 \pm 0.41$ | $64.29 \pm 0.45$ | $81.69 \pm 0.39$ | $88.53 \pm 0.21$ |
| | 1e-4 | $34.88 \pm 0.47$ | $48.66 \pm 0.36$ | $63.15 \pm 0.52$ | $76.73 \pm 8.20$ | $84.42 \pm 0.56$ |

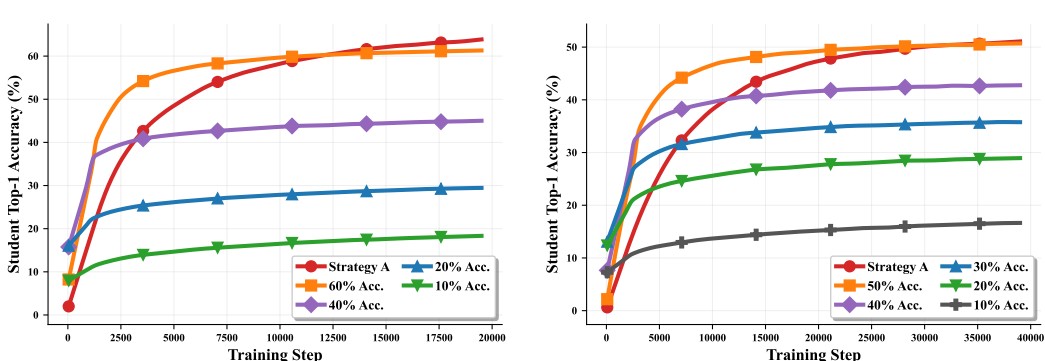

Figure 13: **Comparison of STRATEGY A and STRATEGY B using different teachers on CIFAR100 (Left) and TinyImageNet (Right).** Both show the long-term advantages of STRATEGY A, and the weaker teacher models exhibit more significant improvements in the performance of the student models.

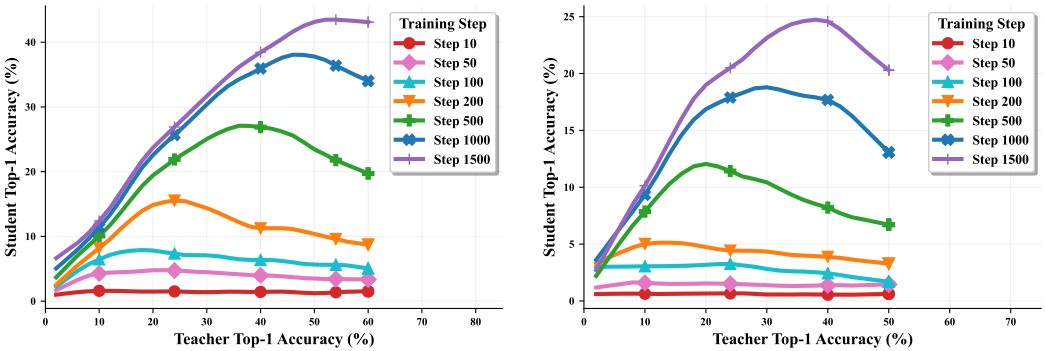

Figure 14: **The different behavior across teachers in different early-stage training steps on CIFAR100 (Left) and TinyImageNet (Right).** For a teacher model with moderate accuracy, there is an advantage in the early stages of training the student model.

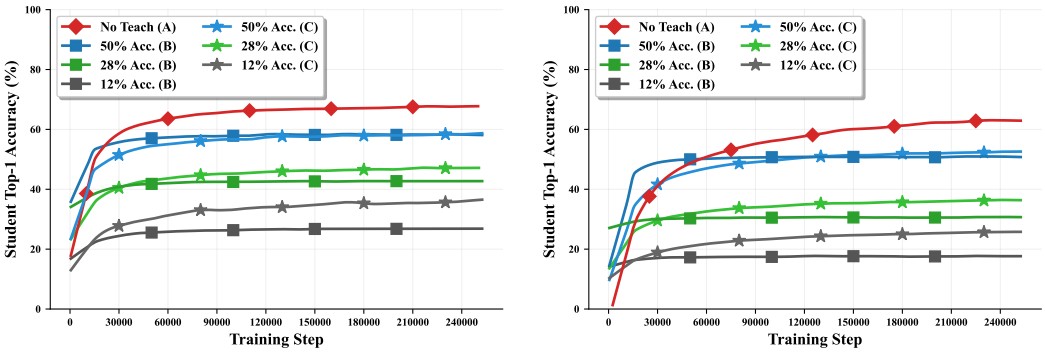

Figure 15: **ResNet50 (Left) and ViT (Right) as backbone architectures on ImageNet.** Under different backbone settings, STRATEGY A shows long-term advantages, while STRATEGY B exhibits short-term benefits. STRATEGY C **effectively addresses the short-term limitations of STRATEGY A and the long-term deficiencies of STRATEGY B**. Moreover, the advantages of STRATEGY C become increasingly prominent when applied to weaker teacher models.

Table 5: **Comparing the student models trained with STRATEGY B and STRATEGY C on different datasets.** Both show that under STRATEGY C, teacher models with moderate or low accuracy are more beneficial for improving the performance of student models.

| Dataset | Step | Strategy | Teacher Top-1 Accuracy (%) ±1% | | | | | | |
|---|---|---|---|---|---|---|---|---|---|
| | | | 2% | 10% | 20% | 30% | 40% | 50% | 60% |
| CIFAR100 | 1950 | B | 7.20 | 12.39 | 23.51 | 31.61 | 36.79 | 42.21 | 43.82 |
| | | C | 5.83 | 13.11 | 22.88 | 27.92 | 31.52 | 34.47 | 35.56 |
| | | Gain | 0.81 | **1.06** | 0.97 | 0.88 | 0.86 | 0.82 | 0.81 |
| | 7800 | B | 12.80 | 16.13 | 27.19 | 35.99 | 43.11 | 51.80 | 58.83 |
| | | C | 9.99 | 19.87 | 31.41 | 37.23 | 42.82 | 48.85 | 53.25 |
| | | Gain | 0.78 | **1.23** | **1.16** | **1.03** | 0.99 | 0.94 | 0.91 |
| | 39000 | B | 17.69 | 20.30 | 31.03 | 38.99 | 45.84 | 55.04 | 62.48 |
| | | C | 18.91 | 32.49 | 39.75 | 44.11 | 49.74 | 56.53 | 61.74 |
| | | Gain | **1.07** | **1.60** | **1.28** | **1.13** | **1.08** | **1.03** | 0.99 |
| Tiny-ImageNet | 3900 | B | 2.63 | 11.82 | 22.31 | 28.15 | 32.48 | 34.68 | - |
| | | C | 2.68 | 11.74 | 20.34 | 24.33 | 26.48 | 27.90 | - |
| | | Gain | **1.02** | 0.99 | 0.91 | 0.86 | 0.82 | 0.81 | - |
| | 15600 | B | 3.92 | 14.76 | 27.13 | 34.21 | 41.20 | 48.55 | - |
| | | C | 4.59 | 17.78 | 28.90 | 34.61 | 39.59 | 44.59 | - |
| | | Gain | **1.17** | **1.21** | **1.07** | **1.01** | 0.96 | 0.92 | - |
| | 78000 | B | 5.63 | 18.62 | 30.07 | 36.56 | 43.60 | 51.38 | - |
| | | C | 8.22 | 24.64 | 35.21 | 41.22 | 46.23 | 51.86 | - |
| | | Gain | **1.46** | **1.32** | **1.17** | **1.13** | **1.06** | **1.01** | - |

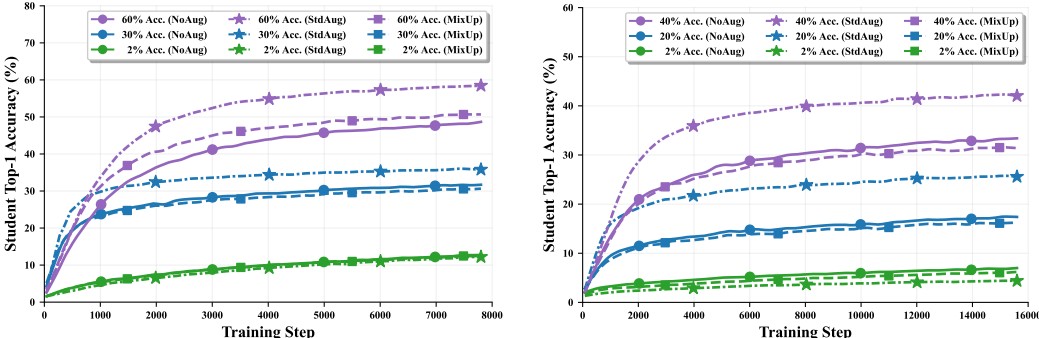

Figure 16: **The different impacts on student model training when applying different image augmentation strategies to the teacher model under STRATEGY B, with the left figure representing CIFAR100 and the right figure representing TinyImageNet.** As the accuracy of the teacher model increases, aggressive data augmentation strategies like MixUp show a stronger advantage. However, when the accuracy of the teacher model is low, MixUp actually exhibits a more pronounced disadvantage.

Table 6: **Comparing the student models trained with STRATEGY B and STRATEGY C on different backbones at different training stages on ImageNet.** Strategy C consistently achieves higher final accuracy than Strategy B across all teacher models.

| Backbone | Step | Strategy | Teacher Top-1 Accuracy (%) ±1% | | |
| --- | --- | --- | --- | --- | --- |
| | | | 12% | 28% | 50% |
| RN50 | 1k | B | 11.21 | 14.47 | 7.25 |
| | | C | 6.74 | 7.96 | 5.98 |
| | | Gain | 0.60 | 0.55 | 0.83 |
| | 50k | B | 25.63 | 41.85 | 56.92 |
| | | C | 29.67 | 43.00 | 54.63 |
| | | Gain | **1.16** | **1.03** | 0.96 |
| | 150k | B | 27.30 | 42.39 | 58.60 |
| | | C | 34.75 | 46.24 | 58.04 |
| | | Gain | **1.27** | **1.09** | 0.99 |
| | 250k | B | 27.00 | 42.64 | 58.02 |
| | | C | 36.05 | 47.71 | 58.60 |
| | | Gain | **1.34** | **1.12** | **1.01** |
| ViT | 1k | B | 7.95 | 6.57 | 4.96 |
| | | C | 5.18 | 5.08 | 4.07 |
| | | Gain | 0.65 | 0.77 | 0.82 |
| | 50k | B | 17.20 | 29.93 | 50.05 |
| | | C | 20.83 | 31.94 | 45.37 |
| | | Gain | **1.21** | **1.07** | 0.91 |
| | 150k | B | 17.70 | 30.34 | 50.98 |
| | | C | 24.68 | 35.10 | 51.66 |
| | | Gain | **1.39** | **1.16** | **1.01** |
| | 250k | B | 17.63 | 30.92 | 51.00 |
| | | C | 25.64 | 36.64 | 52.58 |
| | | Gain | **1.45** | **1.18** | **1.03** |

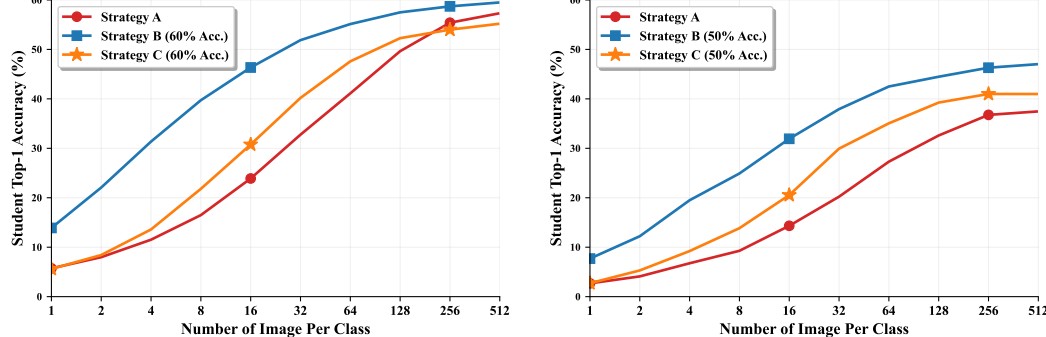

Figure 17: **The scaling behavior using different strategy for training on CIFAR100 (Left) and TinyImageNet (Right).** In both datasets, STRATEGY B consistently shows a greater advantage, STRATEGY C has some advantage when the sample size is small. And for CIFAR100, when the sample size is sufficient enough (i.e., using all samples where IPC=500), the performance of STRATEGY C is eventually surpassed by STRATEGY A.

Table 7: **Comparing the student models trained with STRATEGY A and STRATEGY C at different training stages on ImageNet using ResNet50 as the backbone.** Strategy C significantly accelerates convergence within the first 2.5k steps, regardless of the teacher model used.

| Step | Accuracy | Teacher Top-1 Accuracy (%) $\pm 1\%$ | | | | |
|---|---|---|---|---|---|---|
| | | NoTeach(A) | 12%(C) | 28%(C) | 50%(C) | 62%(C) |
| 0.5k | Stu-Acc. | 0.87 | 3.39 | 3.07 | 2.06 | 2.10 |
| | Gain | – | **3.89** | **3.53** | **2.37** | **2.41** |
| 1k | Stu-Acc. | 2.92 | 6.74 | 7.96 | 5.98 | 5.99 |
| | Gain | – | **2.31** | **2.73** | **2.05** | **2.05** |
| 1.5k | Stu-Acc. | 6.08 | 10.05 | 12.34 | 10.99 | 10.67 |
| | Gain | – | **1.65** | **2.03** | **1.81** | **1.76** |
| 2k | Stu-Acc. | 9.35 | 11.32 | 17.09 | 14.81 | 17.09 |
| | Gain | – | **1.21** | **1.83** | **1.58** | **1.83** |
| 2.5k | Stu-Acc. | 9.86 | 13.42 | 18.57 | 19.89 | 19.94 |
| | Gain | – | **1.36** | **1.88** | **2.02** | **2.02** |
| 3k | Stu-Acc. | 19.15 | 13.50 | 22.14 | 23.13 | 24.59 |
| | Gain | – | 0.70 | **1.16** | **1.21** | **1.28** |
| 3.5k | Stu-Acc. | 20.73 | 14.53 | 22.36 | 26.37 | 27.43 |
| | Gain | – | 0.70 | **1.08** | **1.27** | **1.32** |
| 4k | Stu-Acc. | 23.93 | 14.56 | 24.73 | 24.80 | 28.51 |
| | Gain | – | 0.61 | **1.03** | **1.04** | **1.19** |
| 4.5k | Stu-Acc. | 27.30 | 16.15 | 25.70 | 32.00 | 30.56 |
| | Gain | – | 0.59 | 0.94 | **1.17** | **1.12** |
| 5k | Stu-Acc. | 30.28 | 16.99 | 27.13 | 32.73 | 34.43 |
| | Gain | – | 0.56 | 0.90 | **1.08** | **1.14** |

Table 8: **Comparison of different augmentation method applied for the teacher model using STRATEGY B on different dataset (7,800 steps for CIFAR100 and 15,600 steps for TinyImageNet).** On both datasets, the RandomResizedCrop strategy and the standard data augmentation strategy have a more pronounced effect compared to using the MixUp strategy. And only when the teacher model trained with these aggressive data augmentation strategies has a high accuracy does it provide a greater advantage for the student model, whereas for a weak teacher model, the advantage is not as apparent.

| Dataset | Augmentation (on teacher) | Teacher Top-1 Accuracy (%) ±1% | | | | | |
|---|---|---|---|---|---|---|---|
| | | 2% | 10% | 20% | 30% | 40% | 50% |
| CIFAR100 | NoAug | **12.85** | **16.89** | 24.21 | 31.82 | 38.13 | 46.87 |
| | ColorJitter | 11.09 | 16.65 | **26.75** | 32.65 | 39.65 | 47.64 |
| | Crop | 12.68 | 15.20 | 25.82 | 33.80 | 42.76 | 50.58 |
| | MixUp | 12.64 | 16.56 | 25.19 | 30.78 | 37.51 | 45.59 |
| | CutMix | 9.62 | 14.80 | 22.33 | 29.34 | 36.07 | 43.05 |
| | StdAug | 12.34 | 16.02 | 25.97 | **35.95** | **43.97** | **51.19** |
| TinyImageNet | NoAug | **6.52** | 10.82 | 17.55 | 26.35 | 33.63 | - |
| | ColorJitter | 4.93 | 9.59 | 17.70 | 27.34 | 34.80 | - |
| | Crop | 5.51 | 14.87 | 24.31 | 32.31 | 38.65 | - |
| | MixUp | 6.18 | 8.99 | 16.54 | 25.28 | 31.62 | - |
| | CutMix | 5.62 | 7.40 | 16.48 | 24.07 | 31.62 | - |
| | StdAug | 4.45 | **17.66** | **25.79** | **33.37** | **42.20** | - |

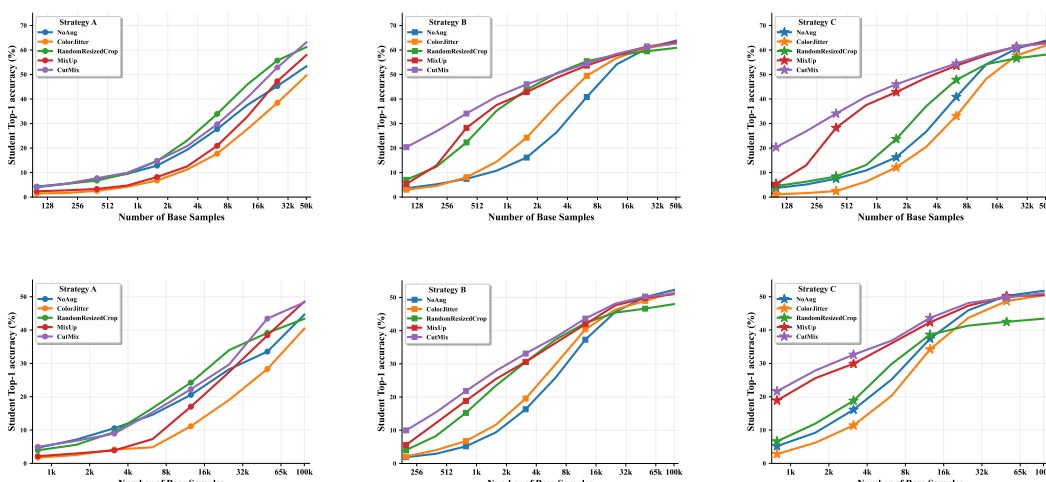

Figure 18: **The scaling behavior of the student model using different image augmentation methods under the three strategies.** The three images above show the results of CIFAR100, while the three images below show the results of TinyImageNet, all of which demonstrate the advantages of STRATEGY B and STRATEGY C compared to the traditional STRATEGY A under limited samples.

