# OpenReview forum: "Equally Critical: Samples, Targets, and Their Mappings in Datasets"
_ICLR.cc/2025/Conference — Submitted to ICLR 2025_

### Official Review · Reviewer_trqj · 2024-10-16

**Soundness:** 3
**Presentation:** 2
**Contribution:** 1
**Rating:** 6
**Confidence:** 1

**Summary:**

The paper investigates the effects of soft target formats, teacher model performance, data quantity, and data augmentation on the training process, within a framework similar to knowledge distillation. The paper primarily presents extensive experiments on CIFAR outputs and summarizes the key findings derived from these experiments.

**Strengths:**

1. Studying neural scaling laws and exploring how to overcome them to achieve a balance between efficiency and performance is both practical and meaningful.
2. The paper presents extensive experimental results from multiple perspectives.

**Weaknesses:**

1. For scaling law research, the experimental scale is too small, even with the so-called “large-scale” Tiny-ImageNet dataset mentioned in the paper.
2. The paper merely lists observations without extracting the underlying insights or potential implications for practical applications.
3. I have concerns about the experimental results. For instance, a ResNet-18 model trained on CIFAR-10 typically achieves over 95% accuracy, yet the best result reported in the paper is only around 80%.
4. The so-called key findings are also trivial. For example, “Soft targets expedite early-stage training”: it’s well-known that knowledge distillation from a teacher model accelerates model convergence. Additionally, “one-hot targets yield superior final accuracy” is unsurprising, as the teacher models in the paper are weak and unable to match the performance of traditional supervised learning, thus hindering final accuracy. I don’t believe this conclusion would hold for more challenging tasks and strong teacher models.
5. I believe the statement “this study is the first to emphasize the critical role of targets in breaking neural scaling power laws” is an overclaim. As mentioned in lines 50-57, there are already existing works on it.
6. The paper does not provide the experimental setup for Figure 1, and the conclusions drawn are inconsistent with those shown in Figure 6(a), where even a 100% subset of soft targets fails to outperform hard targets.
1. Some works regarding offline and online data selection on efficient learning, such as those listed below, should be discussed.

[1] Spanning training progress: Temporal dual-depth scoring (TDDS) for enhanced dataset pruning. CVPR 2024.

[2] Data selection for language models via importance resampling. NeurIPS 2023.

[3] Diversified batch selection for training acceleration. 2024. ICML 2024.

[4] Towards accelerated model training via bayesian data selection. NeurIPS 2023.

[5] Coveragecentric coreset selection for high pruning rates. ICLR 2023.

[6] Beyond neural scaling laws: beating power law scaling via data pruning. NeurIPS 2022.

I’m not sure if I have misunderstood certain parts of the paper. However, based on my current assessment, I believe this paper is not suitable for publication at ICLR. I will adjust my score accordingly, depending on the authors’ clarifications and modifications during the rebuttal phase.

**Questions:**

My questions that need clarification are included in the weaknesses section.


After Rebuttal: I realize that this paper may fall outside my area of expertise. Therefore, I have adjusted my evaluation to a rating of 6 with a confidence level of 1.

---

> ### Author Response · Authors · 2024-11-25
>
> > [W1] For scaling law research, the experimental scale is too small, even with the so-called “large-scale” Tiny-ImageNet dataset mentioned in the paper.
> >
>
> Thank you for your advice. But we want to emphasize that **the scaling law mainly reveals the pattern that model performance improves predictably as a power-law behavior of model size, data size, or compute**, and there are lots of work to reveal that scaling laws are not restricted to large datasets but are also applicable to small datasets.  For example:
>
> - [1] showes in their work that even when dataset sizes are reduced, the relationship between model scale, data size, and performance remains consistent.
> - [2] shows that Vision Transformers adhere to scaling laws on small datasets such as CIFAR10, with performance improvements following similar trends to larger datasets.
> - [3] also take CIFAR10 as a case study and reveal that the pattern observed in CIFAR10 is consistent in ImageNet.
>
> Building on this perspective, our work aligns with these prior studies by showing that meaningful scaling patterns can be derived even from smaller datasets, and here we would like to emphasize that the primary innovative contributions of this work lie in:
>
> - Improving the efficiency of traditional training paradigms by **redefining the mapping between samples and targets**, thereby introducing a novel training framework that, to the best of our knowledge, has not been proposed before.
> - Scaling laws primarily emphasize a power-law relationship, and **case studies on smaller datasets can be meaningfully analyzed** and are sufficient to reveal such patterns.
>
> It is worth noting that in real-world scenarios, ***abundant data is often scarce***. From this perspective, the novel Strategy C we proposed, and the pattern observed in Section 5 provides an effective way to enhance model performance in data-scarce conditions.
>
> To address concerns about scalability, **we have extended our experiments to ImageNet, and we also use ResNet50 and ViT as the backbone for model training**, as shown in Figure 15 and Table 6, 7 for numerical results, Under different backbone settings, we also find that:
>
> - Strategy A shows long-term advantages, while Strategy B exhibits short-term benefits.
> - Our proposed Strategy C **effectively addresses the short-term limitations of Strategy A and the long-term deficiencies of Strategy B**.
> - Moreover, the advantages of Strategy C become **increasingly prominent** when applied to **weaker teacher models**.
>
> We believe this further supports the validity of our findings and their applicability under more settings.
>
> [1] Scaling Laws for Neural Language Models.
>
> [2] Scaling Vision Transformers. CVPR 2022.
>
> [3] Beyond neural scaling laws: beating power law scaling via data pruning. NeurIPS 2022.

---

> > ### Author Response · Authors · 2024-11-25
> >
> > > [W2] The paper merely lists observations without extracting the underlying insights or potential implications for practical applications.
> > >
> >
> > Thank you for your highlighting the importance of theoretical analysis and your concern for potential implications. First, we emphasize that the primary objective of this paper is to explore the **underappreciated yet equally critical roles of targets and mappings** and provide **actionable insights** for deep learning community on how to select **target types, teacher models, and augmentation strategies** to enhance training efficiency. While theoretical explanation is undoubtedly valuable, it is common in the deep learning community for empirical observations to precede and inspire theoretical studies, for example:
> >
> > - Similar to the empirical findings in He's ResNet work in [1], where skip connections were observed to mitigate gradient vanishing issues prior to a detailed theoretical explanation.
> > - Contrastive learning methods, such as [2], were initially evaluated based on empirical results without rigorous theoretical justification.
> > - GPT-3 work[3], where the authors identified and empirically validated scaling phenomena without yet providing a complete theoretical understanding.
> >
> > Thus, we believe our results similarly highlight an important open problem that **merits further exploration**. However, we are willing to provide some **intuitive explanations** here. Specifically:
> >
> > - The effectiveness of model training is, broadly speaking, related to the **quantity of targets**. When fewer targets are present in the short term, the amount of information provided to the model decreases. Among the three proposed strategies, the number of targets from highest to lowest is Strategy B (most), Strategy C (intermediate), and Strategy A (least).
> > - Due to the higher number of targets, **Strategy B can provide more information** in the short term, accelerating model convergence. Our proposed Strategy C can also accelerate short-term convergence since Strategy C has more number of targets than Strategy A.
> > - In the long term, however, **Strategy B generates many incorrect targets**, introducing significant noise that hinders the model from converging to a higher accuracy. Conversely, Strategy A produces only true targets, resulting in less noise and allowing the model to achieve higher final accuracy.
> > - Compared to Strategy B, Strategy C generates fewer targets and thus introduces less noise, **striking a balance between short-term acceleration and long-term convergence**. This enables Strategy C to achieve higher final accuracy than Strategy B, while still retaining some of the short-term benefits.
> >
> > In Section 6, we briefly highlight the **applicability** of our findings
> >
> > - We can extend our findings to larger-scale datasets like text-based tasks. This paper uses image classification **as a case study** to provide a concrete and focused demonstration of the proposed methods, with broader implications left for future exploration.
> > - We would also like to emphasize that this work is **the first to systematically emphasize that a weak teacher model can significantly contribute to the training of student models** in the context of knowledge distillation.
> >
> > Therefore, we believe this work is **a step toward understanding how target mappings impact training dynamics** across domains.
> >
> > [1] Deep Residual Learning for Image Recognition. CVPR 2016.
> >
> > [2] A Simple Framework for Contrastive Learning of Visual Representations. ICML 2020.
> >
> > [3] Language Models Are Few-Shot Learners. NeurIPS 2020.

---

> > > ### Author Response · Authors · 2024-11-25
> > >
> > > > [W3] I have concerns about the experimental results. For instance, a ResNet-18 model trained on CIFAR-10 typically achieves over 95% accuracy, yet the best result reported in the paper is only around 80%.
> > > >
> > >
> > > There are only two experiments where we did not use 90% accuracy for the teacher model and the reason is below:
> > >
> > > - In Section 4.4, when exploring different data augmentation strategies for the teacher model, we did not use a 90% accurate teacher model. This is because the teacher model, when no data augmentation is applied (as in "NoAug" in the paper), has difficulty reaching 90% accuracy, thus **inconvenient to compare with other experimental results**.
> > > - In Section 5.2, when investigating the effect of the number of samples on the student model, the results in Figure 6(a) already show that a teacher model with 80% accuracy has advantages when there are fewer IPCs and disadvantages when there are more IPCs. Apart from that, our supplementary experiment in the Appendix B Figure 10 shows that **the trends observed are consistent** across teacher models with different accuracies.
> > >
> > > For all other experiments, we all used teacher models achieving 90% accuracy, and these results are presented in the paper (section 4.2 -> Figure 3/ Tabel 1, section 4.3 -> Table 2, section 5.3-> Figure 7).
> > >
> > > Additionally, we would like to emphasize that:
> > >
> > > - the main purpose of this paper is to **uncovering key factors influencing scaling laws** that were previously **unaddressed or insufficiently discussed,** by **qualitatively** compare the performance differences of teacher models with varying accuracies. The focus is not on optimizing teacher models using various strategies, as **achieving a highly accurate teacher model is often challenging in practical applications**.
> > > - More importantly, in our experiments, an 80% accurate teacher model is already **sufficiently representative** and capable of revealing the different patterns across teachers.
> > >
> > > > [W4] The so-called key findings are also trivial. For example, “Soft targets expedite early-stage training”: it’s well-known that knowledge distillation from a teacher model accelerates model convergence.
> > > >
> > >
> > > While prior research has indeed shown that soft targets can expedite early-stage training, our work provides **a fresh perspective by analyzing this phenomenon through the lens of sample-to-target mappings**. Our paper introduces STRATEGY C as a compromise between STRATEGIES A and B, and the comparison between A and B just serves as an important motivation for the development of STRATEGY C.
> > >
> > > Furthermore, the insights derived from Figure 3a extend **far beyond** "Soft targets expedite early-stage training." For instance, we emphasize:
> > >
> > > - The **long-term advantages** of Strategy A.
> > > - **Weaker teacher models can significantly enhance student model performance**, even for a randomly initialized teacher model (10% Acc.), it can help the student model to achieve over 30% Acc. To the best of our knowledge, systematic studies focusing on the benefits of weaker teacher models in this specific context are scarce.
> > >
> > > These findings are novel and provide a more comprehensive understanding of the training process.

---

> > > > ### Author Response · Authors · 2024-11-25
> > > >
> > > > > [W5] Additionally, “one-hot targets yield superior final accuracy” is unsurprising, as the teacher models in the paper are weak and unable to match the performance of traditional supervised learning, thus hindering final accuracy. I don’t believe this conclusion would hold for more challenging tasks and strong teacher models.
> > > > >
> > > >
> > > > Intuitively, this observation can be explained as follows:
> > > >
> > > > 1. Since teacher models are trained using STRATEGY A, their accuracy is **inherently lower** than what STRATEGY A can achieve.
> > > > 2. When the teacher model's accuracy is sufficiently high, **the student model's accuracy cannot easily surpass that of the teacher model**.
> > > >
> > > >     For instance, as shown in Figure 3a, with a 90% teacher model, the student model reaches 90% accuracy in only 50 epochs using STRATEGY A, while it takes 100 epochs under STRATEGY B (as shown in Table 2) thus we can see that the student model's accuracy cannot easily surpass that of the teacher model.
> > > >
> > > >
> > > > And we would also like to emphasize that:
> > > >
> > > > - As shown in Figure 3a, we observe that STRATEGY A **does not reach clear convergence** within the 50 epochs shown, whereas a 90% teacher model exhibits noticeable convergence around 7500 steps.
> > > > - It’s worth noting that, as mentioned in [R3], **achieving a very high-accuracy teacher model is challenging in practical applications**. Therefore, the 90%-accuracy teacher model used in Figure 3a is sufficiently representative of real-world conditions.
> > > >
> > > > > [W6] I believe the statement “this study is the first to emphasize the critical role of targets in breaking neural scaling power laws” is an overclaim. As mentioned in lines 50-57, there are already existing works on it.
> > > > >
> > > >
> > > > Lines 50–57 of the paper discuss recent studies highlighting the influence of targets in model training. However, prior work, such as [1], primarily **optimizes targets in the context of unsupervised learning** without systematically exploring **sample-to-target relationships** or conducting experiments on breaking power scaling laws.
> > > >
> > > > In contrast, our paper systematically introduces three distinct target-mapping strategies, and we emphasize that:
> > > >
> > > > - Both STRATEGY B and STRATEGY C can **break the scaling law limitations of the traditional** **STRATEGY A** within a short training period, leading to faster convergence.
> > > > - Furthermore, STRATEGY C can **overcome the scaling law limitations of STRATEGY B** in the long term, resulting in higher final accuracy.
> > > >
> > > >     Our proposed Strategic C **effectively addresses the short-term limitations** of Strategy A and the **long-term deficiencies** of Strategy B.
> > > >
> > > >
> > > > Additionally, our paper is **the first to:**
> > > >
> > > > - systematically investigate the factors influencing the scaling law **from the perspectives of samples (Section 4) and targets (Section 5)**.
> > > > - highlight **the important role that targets generated by a weaker teacher model** can play in guiding the training of the student model throughout the training process.
> > > >
> > > > Thus, we believe this claim is not an overstatement.
> > > >
> > > > [1] Efficiency for Free: Ideal Data Are Transportable Representations. NeurIPS 2024.

---

> > > > > ### Author Response · Authors · 2024-11-25
> > > > >
> > > > > > [W7] The paper does not provide the experimental setup for Figure 1, and the conclusions drawn are inconsistent with those shown in Figure 6(a), where even a 100% subset of soft targets fails to outperform hard targets.
> > > > > >
> > > > >
> > > > > First, we have clearly stated in Claim 5 that Strategy B is **only more effective in low-data** scenarios, and to clarify:
> > > > >
> > > > > - in Figure 1, we use only 10% of the CIFAR-10 training set, i.e., approximately 500 images, which shows that soft targets (Strategy B) are more advantageous than hard targets (Strategy A).
> > > > > - In Figure 6(a), when IPC is around 512, we see that the blue line (representing Strategy B) outperforms the red line (Strategy A), which **aligns** with the conclusions drawn in Figure 1.
> > > > >
> > > > > As for the experimental setup for Figure 1, we have addressed this point in the revised version, specifically on line 715, page 14, where we clarify the number of training steps required for Figure 1. As for other major hyper parameters:
> > > > >
> > > > > - batch size detailed in line 708
> > > > > - learning rate detailed in line 740
> > > > > - training step is set for 10000 steps
> > > > >
> > > > >     The training step keeps the same configuration as in Section 5, since:
> > > > >
> > > > >     - we aimed to ensure consistency across experiments for comparability
> > > > >     - 10000 steps **is sufficient for the model to reach convergence on the small datasets**
> > > > >
> > > > > > [W8] Some works regarding offline and online data selection on efficient learning, such as those listed below, should be discussed.
> > > > > >
> > > > >
> > > > > We appreciate your suggestion to discuss related works on offline and online data selection and acknowledge the importance of sample selection techniques. However, we would like to clarify that **the focus of our paper differs substantially from the methods primarily aimed at sample selection** to improve efficiency. Specifically:
> > > > >
> > > > > - The existing works highlighted mainly concentrate on identifying and selecting subsets of samples to optimize training efficiency. In contrast, our paper emphasizes **the role of labels and the mapping relationships between samples and labels**, which is a fundamentally different perspective.
> > > > > - Our study **systematically analyzes the influence** of target types (e.g., soft vs. one-hot labels) and different sample-to-target mapping strategies on training dynamics and efficiency, **extending far beyond the scope of mere sample selection**.
> > > > > - We provide a **unified loss framework** specifically designed to decouple backbone representation ability from classifier influence, enabling a clearer analysis of the impact of different sample-to-target mappings on training efficiency.
> > > > > - We provide **actionable insights** for deep learning community on how to select **target types, teacher models, and augmentation strategies** to enhance training efficiency.
> > > > >
> > > > >     For instance, Strategy C **effectively addresses the short-term limitations** of Strategy A and the **long-term deficiencies** of Strategy B, and we are the first to systematically emphasize that w**eaker teacher models can significantly enhance student model performance**, a contribution that extends existing literature on knowledge distillation and efficient learning.
> > > > >
> > > > >
> > > > > In summary, while offline and online data selection methods primarily optimize **sample** importance, our study explores the **underappreciated yet equally critical dimensions of targets and mappings**, presenting a complementary perspective in the field of traditional supervised learning. We believe this distinction both validates and strengthens the **novelty and broader impact** of our work.

---

> ### Author Response · Authors · 2024-11-28
> **A Kind Reminder for Reviewer trqj**
>
> **Dear Reviewer trqj,**
>
> Thank you for your detailed feedback and comprehensive review of our paper. Your observations have provided valuable guidance in improving our work, and we have made substantial revisions to address your concerns. Below, we summarize your key points and outline how we have responded:
>
> 1. **Experimental Scale and Applicability of Scaling Laws**:
>     - Your concern: The experimental scale is too small for meaningful scaling law insights.
>     - Our response: We have extended our experiments to include **ImageNet with ResNet50 and ViT backbones**, as shown in Figure 15 and Tables 6 and 7, to validate our findings in larger-scale settings. These results demonstrate consistent **early-stage acceleration and final accuracy improvements** for Strategy C, particularly with **weaker teacher models**, aligning with scaling law patterns observed in prior studies.
> 2. **Depth of Insights and Practical Implications**:
>     - Your concern: Observations lack deeper insights or actionable implications.
>     - Our response: We have provided **intuitive explanations** above for the observed training dynamics, Strategy C balances short-term acceleration (via more targets) and long-term stability (by reducing noise from incorrect targets). Additionally, we discuss practical applications for **selecting target types, teacher models, training strategies** and **augmentation strategies,**  in real-world, data-scarce conditions.
> 3. **Accuracy of Results and Experimental Setups**:
>     - Your concern: ResNet-18 results on CIFAR-10 are subpar, and Figure 1 lacks experimental details.
>     - Our response: We clarified the setup for Figure 1 in the revised version(line 715). The observed ResNet-18 accuracy **aligns well** with our goal of analyzing target mappings. As shown in Appendix B (Figure 10), trends remain consistent across teacher models of varying accuracies.
> 4. **Novelty of Key Findings**:
>     - Your concern: Observations such as "soft targets expedite early-stage training" are trivial.
>     - Our response: While soft targets accelerating convergence is known, our study uniquely **frames this through the lens of sample-to-target mappings**, and there are more non-trivial observations like **weaker teacher models can significantly enhance student model performance**.
> 5. **Relevance of targets to Neural Scaling Laws**:
>     - Your concern: The claim of ”the first to emphasize the critical role of targets in breaking scaling laws“ is overstated.
>     - Our response: Our claim focuses on **systematically exploring sample-to-target mappings** and demonstrating their role in improving training efficiency across scales. Strategy C, in particular, mitigates the inefficiencies of traditional approaches, which has ***NEVER BEEN PROPOSED before***.
> 6. **Additional Literature on Data Selection**:
>     - Your concern: Relevant works on offline and online data selection are not discussed.
>     - Our response: We acknowledge these studies and clarified that our focus is ***FUNDAMENTALLY DIFFERENT***. While data selection methods optimize sample importance, our work emphasizes the role of ***TARGETS AND THEIR MAPPINGS***, presenting a complementary perspective that broadens the understanding of efficient learning.
>
> We hope that our clarifications and revisions adequately address your concerns. If you find the updates align with your expectations, we would be grateful if you could consider revising your score, as it would significantly support the progression of our work.
>
> Please let us know if you have any further feedback or questions. Thank you again for your thoughtful review and for engaging with our submission.
>
> **Best regards,**
>
> The Authors of Submission 10518

---

> > ### Author Response · Authors · 2024-11-30
> > **Kindly Awaiting for Reviewer trqj’s Feedback**
> >
> > **Dear Reviewer trqj,**
> >
> > We hope this message finds you well. Your feedback has been invaluable in refining our work, and we have made every effort to address your concerns, including **extending experiments** to larger datasets, providing **deeper insights** into observed phenomena, **clarifying experimental setups** and the **main contribution of our paper.**
> >
> >  Following up on our previous reminder, if the clarifications and revisions resolve your concerns, we would greatly appreciate it if you could consider revising your score. Your updated evaluation would be tremendously helpful for the progression of our work. We remain fully available to engage further and address any remaining questions.
> >
> > Thank you again for your time and contributions to this process.
> >
> > **Best regards,**
> >
> > The Authors of Submission 10518

---

> > > ### Author Response · Authors · 2024-12-01
> > > **Eagerly Anticipating Reviewer trqj’s Feedback**
> > >
> > > **Dear Reviewer trqj,**
> > >
> > > Thank you very much for your detailed review and valuable suggestions. With the discussion phase nearing its conclusion (December 2nd), we wanted to follow up to ask if our responses have clarified your concerns. Should you have any further questions or comments, please let us know, we would be glad to address them.
> > >
> > > We deeply appreciate your time and effort in helping improve our submission!
> > >
> > > **Best regards,**
> > >
> > > The Authors of Submission 10518

---

> > > > ### Comment · Reviewer_trqj · 2024-12-03
> > > >
> > > > Sorry for the delayed response, and thank you for your detailed reply. I carefully reviewed the other reviewers’ comments and your rebuttal in an effort to understand the core contributions and insights of this work. However, I found this challenging, likely because the work is somewhat outside my area of expertise. To avoid potential misjudgment, I have decided to adjust my evaluation to a rating of 6 with a confidence level of 1.

---

> ### Author Response · Authors · 2024-12-03
>
> **Dear Reviewer trqj,**
>
> Thank you for your thoughtful response and for carefully reviewing the other reviewers’ comments and our rebuttal. We appreciate your decision to adjust your evaluation and your careful consideration of our work.
>
> Thank you again for your time and feedback.
>
> **Best regards,**
>
> The Authors of Submission 10518

---

### Official Review · Reviewer_pm2n · 2024-11-03

**Soundness:** 2
**Presentation:** 2
**Contribution:** 1
**Rating:** 3
**Confidence:** 3

**Summary:**

The authors study how three different categories of data augmentation (dubbed Strategies A, B, and C) affect the rate of improvement in model test accuracy with respect to training budget and training dataset size. The methods are grouped by how the data augmentation affects the sample *label* rather than the features: the first group uses standard one-hot labels, the second uses soft labels from a teacher model but recomputed on each augmented feature vector, and the final group computes a soft label only on the pre-augmented feature vector and uses this for all augmented samples. They conduct experiments to determine which augmentation strategy is optimal in different settings and provide general recommendations based on these results.

**Strengths:**

Data and computational efficiency are highly relevant practical problems. As we reach fundamental upper limits on the possible size of training datasets, finding ways to improve the neural scaling laws that have been observed until now will be essential for continuing to improve model capabilities. Thus, the stated problem under study is relevant to the ICLR community.

**Weaknesses:**

The main thrust of the paper is that, in the context of improving neural scaling laws, their "finding underscores the significance of the exploration the target component, a frequently overlooked aspect in the deep learning community." The discussion of neural scaling laws centers of the fact that exponentially larger datasets are needed to achieve only marginal performance improvements; in particular, these scaling laws are a problem only once we have reached the "extremely large dataset regime." On the other hand, the insights the authors provide showing the advantage of augmented targets all occur in low data regimes. As they return to the size of the full CIFAR-10 dataset, regular 1-hot labels have the best performance. Thus, it's unclear what relevance the insights in the paper have to the stated practical problem of interest, i.e., datasets at a scale far beyond that of CIFAR-10.

Related to this first point, Claim 3 on the efficacy of the different strategies is misleading. In particular, the best overall performance is in fact obtained by Strategy B with a 90% accurate teacher model (Table 2 in the appendix).

The choice to separate the model training into the "backbone" (feature extractor) and classifier is motivated by the claim that the cross-entropy loss cannot handle soft labels, but this is not true. The cross-entropy can be computed between any two discrete distributions with the same support (https://en.wikipedia.org/wiki/Cross-entropy). In fact, the KL divergence and cross-entropy differ by a quantity which is constant w.r.t. the trained model, so the gradients for the proposed training strategy for the backbone are the same as the standard CE loss. It should also be noted that previous data augmentation methods which use soft labels (such as MixUp) also apply CE with the soft labels directly.

The description of the results in Fig. 5a seems to be incorrect. This paragraph states that MixUp > standard augmentation > no augmentation, but the plot has standard augmentation with the highest performance with MixUp and no augmentation approximately equal for the relevant purple and blue lines.

Minor: A more descriptive name than "Strategy A/B/C" would make it easier for the reader to remember the salient features of the different augmentation methods.

**Questions:**

Can the authors explain why the results/insights from the paper are relevant to neural scaling laws at the practically relevant scales discussed in the introduction?

---

> ### Author Response · Authors · 2024-11-25
>
> > [W1]  The main thrust of the paper is that, in the context of improving neural scaling laws, their "finding underscores the significance of the exploration the target component, a frequently overlooked aspect in the deep learning community." The discussion of neural scaling laws centers of the fact that exponentially larger datasets are needed to achieve only marginal performance improvements; in particular, these scaling laws are a problem only once we have reached the "extremely large dataset regime."
> >
>
> We appreciate your observations and would like to clarify that:
>
> - Neural scaling laws mainly describe how model performance follows a power-law relationship with compute, data size, and model scale, **emphasizing the inefficiency of performance improvements** as these factors scale.
> - The main goal of our paper is to explore methods to **mitigate this inefficiency** during training.
>
> And we also would like to emphasize that the primary innovative contributions of this work lie in:
>
> - Improving the efficiency of traditional training paradigms by **redefining the mapping between samples and targets**, thereby introducing a **meaningful and** **novel training framework** that, to the best of our knowledge, has never been proposed before.
> - Scaling laws primarily emphasize a power-law relationship, and **case studies on relatively larger datasets (like TinyImageNet) can be meaningfully analyzed** and are sufficient to reveal such patterns.
>
> The research focus on targets presented in our paper is **not contradictory to neural scaling laws**. To provide a quick recap, we summarize how **modifying the mapping between samples and targets can influence model training efficiency**:
>
> - Strategy B **accelerates early convergence**, achieving better model performance with fewer computational resources compared to traditional methods (*Figure 3a*).
> - **Weaker teacher models can also aid early convergence**, providing better performance with less computing than traditional methods (*Figure 3b*).
> - Our proposed Strategy C **effectively addresses the short-term limitations of Strategy A and the long-term deficiencies of Strategy B,** achieving higher model performance improvements for the same computational resources (*Figure 4*).

---

> ### Author Response · Authors · 2024-11-25
>
> > [W2] On the other hand, the insights the authors provide showing the advantage of augmented targets all occur in low data regimes. As they return to the size of the full CIFAR-10 dataset, regular 1-hot labels have the best performance. Thus, it's unclear what relevance the insights in the paper have to the stated practical problem of interest, i.e., datasets at a scale far beyond that of CIFAR-10.
> >
>
> Thank you for your concern about our insights. Here we would like to emphasize that:
>
> - In most real-world deep learning scenarios, **abundant data is often scarce.** Modern deep learning research increasingly focuses on achieving better performance **under limited data conditions** [1,2], and our findings are **highly relevant to practical applications**.
> - Take Section 5.2 for example, different data augmentation strategies affect model performance differently. Under limited samples, **specific data augmentation strategies help the model converge more effectively**, breaking the traditional power-law relationship between data size and model performance.
>
> Regarding your observation that "regular 1-hot labels have the best performance with the full CIFAR-10 dataset," we provide two clarifications:
>
> - In practical scenarios, it is often difficult to determine whether the available dataset is "sufficient." Many real-world applications involve insufficient data, where **different training strategies can offer significant advantages** [3,4].
> - The "No Free Lunch" theorem [5] reminds us that no single machine learning method is optimal for all situations🧐. In real-world settings, the choice of method depends on the specific conditions. In the context of our findings:
>     - When data is **limited**, **Strategy B** is preferable.
>     - When data is **abundant**, **Strategy A** performs best.
>
> To address concerns about scalability, **we have extended our experiments to ImageNet, and we also use ResNet50 and ViT as the backbone for model training**, as shown in Figure 15 and Table 6,7 for numerical results, Under different backbone settings, we also find that
>
> - Strategy A shows long-term advantages, while Strategy B exhibits short-term benefits.
> - Our proposed Strategy C **effectively addresses the short-term limitations of Strategy A and the long-term deficiencies of Strategy B**.
> - Moreover, the advantages of Strategy C become **increasingly prominent when applied to weaker teacher models**.
>
> We believe this further supports the validity of our findings and their applicability under more settings.
>
>
>
> [1] Beyond neural scaling laws: beating power law scaling via data pruning. NeurIPS 2022.
>
> [2] Deep learning on a data diet: Finding important examples early in training. NeurIPS 2021.
>
> [3] Scaling laws of synthetic images for model training... for now.  IEEE 2024.
>
> [4] How much data are augmentations worth? an investigation into scaling laws, invariance, and implicit regularization. ICLR 2023.
>
> [5] No Free Lunch Theorems for Optimization. IEEE 1997.

---

> ### Author Response · Authors · 2024-11-25
>
> > [W3] Claim 3 on the efficacy of the different strategies is misleading. In particular, the best overall performance is in fact obtained by Strategy B with a 90% accurate teacher model (Table 2 in the appendix).
> >
>
> We acknowledge that the description of Claim 3 could have been clearer. However, as shown in Table 2 in the appendix, our results indicate that:
>
> - When the teacher model's accuracy is fixed, the gain from Strategy C **continues to increase as training progresses**. For example, with a 90% accurate teacher model, the gain from Strategy C progresses from 0.94 → 0.96 → 0.99, showing a steady **improvement over time**.
> - Additionally, Figure 8 presents the accuracy of the student model trained with an 80% accurate teacher model over different training steps. From the plot, we observe that **Strategy C does not show a significant convergence bottleneck**, and the gap between Strategy C and Strategy B grows as training progresses.
>
> Moreover, we also would like to give an intuitive explanation to why **the performance improvement of Strategy C over Strategy B is positively correlated with the number of training steps** and **Strategy C can achieve higher final accuracy than Strategy B**. As noted in line 344 of the paper, Strategy C generates fewer distinct targets during training compared to Strategy B. For instance, on CIFAR10 with 100 training epochs, the ratios of sample-to-target mappings are as follows:
>
> - **Strategy A:** 50,000 × 100:10 = 500,000:1
> - **Strategy B:** 50,000 × 100:50,000 = 1:1
> - **Strategy C:** 50,000 × 100:50,000 = 100:1
>
> Strategy C establishes a mapping between samples and targets that **scales proportionally with the number of training epochs**. Consequently, when the number of training steps is sufficiently large, Strategy C will **asymptotically approach the performance of Strategy A**.
>
> Based on these observations, we have strong reason to believe that, **with further training, Strategy C would achieve the highest final accuracy**.
>
> > [W4] The choice to separate the model training into the "backbone" (feature extractor) and classifier is motivated by the claim that the cross-entropy loss cannot handle soft labels, but this is not true. The cross-entropy can be computed between any two discrete distributions with the same support (https://en.wikipedia.org/wiki/Cross-entropy). In fact, the KL divergence and cross-entropy differ by a quantity which is constant w.r.t. the trained model, so the gradients for the proposed training strategy for the backbone are the same as the standard CE loss. It should also be noted that previous data augmentation methods which use soft labels (such as MixUp) also apply CE with the soft labels directly.
> >
>
> The primary reason for this decision is not solely due to the claim that cross-entropy loss cannot handle soft labels, as you pointed out. Instead, the motivation is twofold:
>
> - In traditional knowledge distillation, where the teacher model’s backbone and classifier are trained jointly, there can be cases where **the backbone extracts strong features, but the classifier’s performance does not align well with the teacher’s output**. This can lead to suboptimal training outcomes.
> - By decoupling the backbone from the classifier, which aligns with approaches in unsupervised learning where feature extraction is performed independently of specific downstream tasks [1,2], we can **better assess and refine the feature extraction capacity of the model**, ensuring that both components perform optimally.
>
> We hope this clarification better conveys our rationale for this approach. Thank you for highlighting this.
>
> [1] A Simple Framework for Contrastive Learning of Visual Representations. ICML 2020.
>
> [2] Bootstrap Your Own Latent: A New Approach to Self-Supervised Learning. NeurIPS 2020.
>
> > [W5] The description of the results in Fig. 5a seems to be incorrect. This paragraph states that MixUp > standard augmentation > no augmentation, but the plot has standard augmentation with the highest performance with MixUp and no augmentation approximately equal for the relevant purple and blue lines.
> >
>
> We acknowledge the mistake in our description in line 404. As you correctly pointed out, the results should be interpreted as: standard augmentation > MixUp > no augmentation.  However, we would also like to emphasis that
>
> - *For high-accuracy teacher models, applying MixUp augmentation to the teacher does not significantly benefit the student model’s training and may even perform worse than using a NoAug-trained teacher model.*
>
> which is also an intriguing and novel observation😊 we have corrected this misstatement in the revised version.

---

> ### Author Response · Authors · 2024-11-25
>
> > [Q1] Can the authors explain why the results/insights from the paper are relevant to neural scaling laws at the practically relevant scales discussed in the introduction?
> >
>
> Thank you for your question. In addition to systematically introducing the three distinct target-mapping strategies, we would also like to emphasize that:
>
> - Both STRATEGY B and STRATEGY C can **break the scaling law limitations of the traditional** **STRATEGY A** within a short training period, leading to faster convergence.
> - Furthermore, STRATEGY C can **overcome the scaling law limitations of STRATEGY B** in the long term, resulting in higher final accuracy.
>
>     Our proposed Strategic C **effectively addresses the short-term limitations** of Strategy A and the **long-term deficiencies** of Strategy B.
>
>
> Additionally, our paper is **the first to:**
>
> - systematically investigate the factors influencing the scaling law **from the perspectives of samples (Section 4) and targets (Section 5)**.
> - highlight the important role that targets generated by a **weaker teacher** model can play in guiding the training of the student model throughout the training process **in the context of knowledge distillation**.
>
> In Section 6, we briefly highlight the **applicability** of our findings:
>
> - We can extend our findings to larger-scale datasets like text-based tasks. This paper uses image classification **as a case study** to provide a concrete and focused demonstration of the proposed methods, with broader implications left for future exploration.
> - We believe this work is **a step toward understanding how target mappings impact training dynamics** across domains.
>
> And  we have also **extended our experiments to ImageNet** and we also use **ResNet50 and ViT** as backbone for model training to verify the effectiveness of our proposed Strategy. Thus, we believe this work is highly  relevant to neural scaling laws at the practically relevant scales.

---

> > ### Comment · Reviewer_pm2n · 2024-11-26
> >
> > I have read the author response and looked at the revised paper.
> >
> > The additional larger-scale experiment on ImageNet is appreciated, but unfortunately it is a negative result. The authors have emphasized that the proposed Strategy C retains both the short-term benefits of Strategy B and the long-term benefits of Strategy A, but I believe the results of this plot are more accurately interpreted as retaining the downsides of the two methods. In particular, in Fig. 15, for all teacher accuracies, Strategy C is completely dominated by the max of Strategies A and B for both ResNet50 and the ViT.
> >
> > Regarding the main motivation of the paper with respect to neural scaling laws, as the authors themselves state:
> >
> > >Neural scaling laws mainly describe how model performance follows a power-law relationship with compute, data size, and model scale, emphasizing the inefficiency of performance improvements **as these factors scale.** [Emphasis mine.]
> >
> > The scenarios in this paper in which some benefit is observed is when both data and compute are small-scale, whereas when either compute (number of training steps) or dataset size becomes large, the insights are not applicable. Breaking through neural scaling laws should give some benefit in at least one of the two tails of data or compute, so the main concern about the relevance of the paper to practical deep learning scenarios is still an issue. (As a side note, in all of the plots, the *starting* accuracy (training step 0) of Strategy A is lower than most of the other methods, which accounts for at least some of the early lag. It's unclear why the different strategies should have different starting points.)
> >
> > For these reasons, I will retain my score.

---

> ### Author Response · Authors · 2024-11-27
>
> Thank you for your thoughtful comments and for carefully reviewing our revised paper. We have further updated the paper to include additional results, specifically adding Table 7, which provides a detailed comparison of **early-stage numerical results between Strategies A and C**. And we address your concerns as follows:
>
> **1. On the starting accuracy discrepancy in Figure 15 and other plots:**
>
> Here, we provide clarifications for the observed differences:
>
> - **Validation frequency:** For large-scale datasets like ImageNet, performing validation after every training step is computationally infeasible. Instead, we evaluate every 500 steps. This does not affect the trends observed, as shown in Table 6,7, where we provide further comparisons that confirm consistency in observed patterns.
> - **LOWESS smoothing:** As noted in line 734 of the paper, we used the LOWESS method for local smoothing to reduce noise and better illustrate overall trends. Because this approach incorporates data from neighboring points, and since Strategies B and C **accelerate early-stage convergence significantly**, the smoothed curves may visually imply that these methods achieve high accuracy almost immediately. And we believe this visualization highlights the **early advantage** of Strategies B and C compared to the **slower convergence** of Strategy A.
>
>     Recognizing that smoothing may influence the interpretation of early-stage results, we addressed this concern in line 736 by providing **concrete numerical results** in Tables 2, 5, 6, 7.
>
>
> In Table 7, we further provide detailed **comparisons between Strategies A and C in early training steps**. These results show that Strategy C significantly **accelerates convergence** within the first 2k steps, regardless of the teacher model used. For comparisons between Strategies B and C, detailed results are already provided in Table 6.
>
> **2. On Strategy C’s effectiveness and the interpretation of results in Figure 15:**
>
> We respectfully disagree with your interpretation that Strategy C "retains the downsides" of Strategies A and B. Allow us to clarify:
>
> - **Early-stage acceleration:** As mentioned above and supported by Table 7, Strategy C demonstrates **substantial early-stage acceleration across all teacher models**.
> - **Final accuracy:** Contrary to your claim, the results in Figure 15 and Table 6 clearly show that Strategy C consistently achieves **higher final accuracy than Strategy B** across all teacher models. The numerical gains (in Table 6) for Strategy C compared to Strategy B, particularly with weaker teacher models, are evident and significant.
> - **Why Strategy C works:** To aid understanding, we have previously provided an intuitive explanation: Strategy C represents a novel compromise between Strategies A and B. By reducing the **sample-to-target ratio** compared to Strategy A, Strategy C naturally **accelerates early convergence**. Additionally, by increasing sample-to-target ratio, Strategy C effectively **decreases the noise inherent** in Strategy B and improves final accuracy.
>
>     We believe this dual benefit distinguishes Strategy C and validates its utility as a **meaningful contribution compare to the traditional training paradigm**.
>
>
> **3. On the relevance of findings to large-scale datasets and neural scaling laws:**
>
> As for the practical deep learning scenarios, we clarify that:
>
> - **Insights remain valid:** Even with a **large dataset** like ImageNet, as **training steps** increase, we observe consistent patterns that **align with our conclusions in the paper** from relatively larger scale experiments (like TinyImageNet). Strategy C continues to demonstrate **early-stage acceleration and final accuracy gains**, particularly with weaker teacher models.
> - **Core contribution:** The main contribution of our work is introducing a **novel perspective on the sample-target mapping** and systematically exploring its impact on training dynamics. This perspective offers a meaningful approach to **addressing inefficiencies in neural network training, which is relevant across data and compute scales in neural scaling law**.
>
> ---
>
> Apart from that, as you correctly noted in your Strengths section:
>
> - *"Finding ways to improve the neural scaling laws that have been observed until now will be essential for continuing to improve model capabilities. Thus, the stated problem under study is relevant to the ICLR community."*
>
> We firmly believe our work is a **meaningful step** toward this goal and **merits further exploration**. By introducing new insights into how target mappings influence training dynamics, we propose **practical methods** for deep learning community on how to select **target types, teacher models, and augmentation strategies** to enhance training efficiency.

---

> ### Author Response · Authors · 2024-11-28
> **A Kind Reminder for Reviewer pm2n**
>
> **Dear Reviewer pm2n,**
>
> Thank you for your thorough review and detailed feedback on our paper. Your insights have been instrumental in refining our work, and we sincerely appreciate the time and effort you’ve dedicated to assessing our submission. Below, we summarize your key concerns and outline how we’ve addressed them in the revised version:
>
> 1. **Effectiveness of Strategy C**:
>     - Your concern: Strategy C seems to combine the drawbacks of Strategies A and B, without retaining their respective advantages.
>     - Our response: We want to emphasis that:
>         - The main purpose of our paper is ***NOT*** to propose a **universal SOTA** or even an ***OMNISCIENT*** strategy that consistently achieves the best all the time, but to examine traditional training paradigms through the lens of the ***MAPPING*** relationship between samples and targets.
>         - It is ***UNREALISTIC IN PRACTICAL SCENARIOS to expect a single algorithm to perform OPTIMALLY THROUGHOUT THE ENTIRE TRAINING PROCESS***.
>
>         We believe that Strategy C provides **improvements over traditional Strategy A** in the short term and outperforms Strategy B in the long term, offering a ***NOVEL PERSPECTIVE*** for addressing training inefficiencies, and we also emphasis the important role a **weaker model** can play. Additionally, we provided an **intuitive explanation** of Strategy C's scaling advantage through its sample-to-target ratio adjustments.
> 2. **Relevance to Neural Scaling Laws**:
>     - Your concern: The paper's findings appear relevant only in low-data regimes, with diminishing applicability as data size or compute increases, making them less impactful for practical neural scaling scenarios.
>     - Our response: We clarified that our insights provide **actionable insights** for deep learning community on how to select **target types, teacher models, and augmentation strategies** to **mitigate inefficiencies in training dynamics** which are prevalent in real-world applications, we extended our experiments to ImageNet using ResNet50 and ViT backbones, providing new evidence supporting Strategy C’s utility in both **early-stage acceleration and final accuracy gains**.
>
> We hope that these updates adequately address your concerns. If our clarifications and revisions meet your expectations, we kindly request you to consider revising your score, as this would greatly support the progression of our work.
>
> Should you have any additional feedback or questions, we remain available to further discuss. Thank you again for your invaluable contributions and for engaging deeply with our submission.
>
> **Best regards,**
>
> The Authors of Submission 10518

---

> ### Author Response · Authors · 2024-11-30
> **Kindly Awaiting for Reviewer pm2n’s Feedback**
>
> **Dear Reviewer pm2n,**
>
> Thank you again for your feedback on our paper. We have carefully addressed your concerns about the effectiveness of Strategy C in practical large-scale settings and we wish to re-emphasize that our primary objective is to
>
> - ***SYSTEMATICALLY*** analyse the ***CRITICAL*** and often ***OVERLOOKED*** role of ***TARGETS AND THEIR MAPPINGS*** in a dataset, thus propose and validate a ***NOVEL, MEANINGFUL*** and ***IMPACTFUL*** training paradigm that benefits ***EFFICIENT LEARNING.***
> - Provide ***ACTIONABLE INSIGHTS*** for deep learning community on how to select ***TARGET TYPES, TEACHER MODELS***, and ***AUGMENTATION STRATEGIES*** to enhance training efficiency and address key challenges in neural scaling laws.
>
> We respect your thoughtful suggestions, but directly reject our paper may potentially overlook the substantial and novel contributions that it brings to the field. We firmly believe that the **significance of understanding target mappings** is a fundamental topic that **deserves attention and further exploration**, and our paper is a vital step in this direction.
>
> We believe our paper delivers these insights while addressing all points raised in your review and respectfully request that you **reconsider your score** based on our clear clarification. If there are any remaining concerns, we are ready to provide further clarifications promptly.
>
> Thank you again for your time and for contributing to this process.
>
> **Best regards,**
>
> The Authors of Submission 10518

---

> > ### Author Response · Authors · 2024-12-01
> > **Eagerly Anticipating Reviewer pm2n’s Feedback**
> >
> > **Dear Reviewer pm2n,**
> >
> > We greatly appreciate your time and effort in reviewing our submission and providing constructive feedback. As the discussion deadline (December 2nd) is approaching, we would like to kindly ask if our responses have resolved your concerns. Please let us know if you have any additional questions or comments, we would be happy to engage further.
> >
> > Thank you again for your thoughtful contributions!
> >
> > **Best regards,**
> >
> > The Authors of Submission 10518

---

> > > ### Comment · Reviewer_pm2n · 2024-12-02
> > >
> > > >Final accuracy: Contrary to your claim, the results in Figure 15 and Table 6 clearly show that Strategy C consistently achieves higher final accuracy than Strategy B across all teacher models. The numerical gains (in Table 6) for Strategy C compared to Strategy B, particularly with weaker teacher models, are evident and significant.
> > >
> > > The exact words of my claim were "In particular, in Fig. 15, for all teacher accuracies, Strategy C is completely dominated by **the max** of Strategies A and B for both ResNet50 and the ViT." It is true that Strategy C has better final accuracy than Strategy B, but both final accuracies are worse than Strategy A which does not require any teacher model or modification to standard training procedures. While it may be "**UNREALISTIC IN PRACTICAL SCENARIOS to expect a single algorithm to perform OPTIMALLY THROUGHOUT THE ENTIRE TRAINING PROCESS,**" it is entirely reasonable to expect a newly proposed method to obtain improvement at *some* point in the training process, but at no point during training would it be preferable to use Strategy C over the better of Strategies A & B. I retain my score.

---

> ### Author Response · Authors · 2024-12-03
> **Clarification on The Motivation**
>
> **Dear Reviewer pm2n,**
>
> Thank you for your thorough review and for engaging deeply with our work throughout the discussion phase. We greatly appreciate your thoughtful feedback and understand your perspective.
>
> From the strategies discussed in our paper, we think it is possible to hypothesize a hybrid approach: employing Strategy B during the early stages and then switching to Strategy A in the later stages to benefit from its superior final accuracy, may obtain improvement at *some* point in the training process and could potentially yield a more efficient training paradigm.
>
> **HOWEVER**, we deliberately chose not to pursue this direction because it does **not align** with our primary research objective. Our paper is ***NOT*** intended to propose a SOTA method. Instead, as highlighted in our title, we focus on the **critical role of samples, targets, and their mappings.** Our insights extend far beyond Strategy C itself, but more importantly, we offer ***ACTIONABLE*** guidance on selecting targets, teacher models, and augmentations to enhance training efficiency.
>
> Please allow us to kindly suggest that a decision to directly reject our work without a more **comprehensive** consideration of its **novelty and practicality** might overlook its broader value and insights it provides to the community. While we respect your final decision, we hope you will reconsider it in light of the fundamental **motivations and contributions** of our paper.
>
> Thank you again for your time, thoughtful engagement, and constructive feedback throughout this process.
>
> **Best regards**,
>
> The Authors of Submission 10518

---

### Official Review · Reviewer_apkJ · 2024-11-04

**Soundness:** 2
**Presentation:** 2
**Contribution:** 2
**Rating:** 5
**Confidence:** 3

**Summary:**

The main goal of the paper is to investigate the influence of different target encodings on learning efficiency in an exploratory way that generates some interesting new hypotheses.
The authors categorize three types of target encodings: hard labels (A), soft labels based on the un-augmented input (B) and soft labels from the augmented input (C).
To empirically compare the three approaches, they define a teacher-student training setup.
The teacher is trained using approach A and is used to generate soft labels for approaches B and C. Student networks are then trained using approaches A, B and C and they are evaluated with respect to top-1 accuracy on three image classification tasks. The network structure consists of a backbone and two heads, one for the soft labels and one for the hard labels respectively. For approaches B and C, the hard labels are only used to train the network head for the hard labels while the backbone and head for the soft labels are trained using the soft labels. As the authors are interested in disentangling the influence of the soft and hard labels on the representational capacity of the backbone, this training setup prevents the hard labels from influencing the structure of the backbone. In the remainder of the paper they evaluate the three methods in different experimental setups, where they use teachers with different quality, select different numbers of observations per class and also vary the data augmentation schemes for the teacher and the student. From these experimental results they are able to derive six findings that relate the varied experimental factors to the accuracy of the student model. For example, soft targets can speed up early training and student networks – when training using approach B and C – are limited by the capacity of the teacher.

**Strengths:**

They pose an interesting question how different target encodings influence training efficiency of neural networks
Interesting experiments are designed that investigate questions such as how the quality of labels affects the accuracy of the student during different stages of the training, whether better teacher performance always entails better student performance or the interplay of data augmentation for the student and the teacher.
All experiments are repeated at least five times.
The paper is well structured and easy to read

**Weaknesses:**

The experiments fail to consider other possibly relevant factors. For example, it is possible that strategy A in the results from figure 3a) simply needs a different learning rate
While the experiments are repeated at least five times, no uncertainty quantification (such as standard error) is included in the plots or the analysis.
Research data:
No code is provided
Experiment results are not included, e.g. as csv files
While interesting experiments are designed and phenomena are observed, little explanation is offered as to why these patterns are being observed.
For example, in the context of Figure 7 it would be interesting to discuss the effects the different augmentation strategies have on the class probabilities, which might explain why some augmentation methods perform worse with strategy C than with strategy B.
When an augmentation method actually changes the class probability of an image (such as random cropping), using strategy C to train the network does not reduce noise but instead feeds the network wrong soft labels.

**Questions:**

Figure 1: How are the results obtained? For example, how were the number of epochs determined?
Section 3.2: I do not fully understand the motivation behind the design of a unified loss function. It is said that CE is unable to exploit information from soft targets, but CE can be used with soft targets, so it’s not clear to me what is meant with that statement. Also, why is it not a valid evaluation strategy to evaluate strategies B and C by training the network exclusively using the soft labels with cross-entropy?
Later it is said that KL divergence is used to leverage the information in the soft targets, but this seems to contradict what was said above, i.e. that CE cannot make use of soft labels, but minimizing CE and KL divergence is equivalent.
Figure 5:
Which strategy was used to train the student?
On line 405 it says: “MixUp-trained teacher models achieve superior performance compared to standard augmentation”, but this is not supported by Figure 5a?
Why are the results of the experiments from the appendix not summarized in the main paper, i.e. whether they support the six key findings?

---

> ### Author Response · Authors · 2024-11-25
>
> > [W1]  it is possible that strategy A in the results from figure 3a) simply needs a different learning rate. While the experiments are repeated at least five times, no uncertainty quantification (such as standard error) is included in the plots or the analysis
> >
>
> Thank you for your advice. As for your concern in Strategy A from Figure 3a, in our revised version, we further show that **tuning a different learning rates and batch size cannot help**, as shown in Figure12 (line 955, page 18), all results support our finding in Claim 1 that:
>
> - Strategy A shows long-term advantages, while Strategy B exhibits short-term benefits.
> - The **important role** that targets generated by a **weaker teacher** model can play in guiding the training of the student model throughout the training process.
> - For simplicity, we compare Strategy A and B as a case study and provide numerical results in Table 4 (line 95 on page 18) where we experiment **five times** and record the final accuracy with **standard error** using different teacher models. The experimental results demonstrate that our error margins are **sufficiently small**, ensuring the reliability of our **reported findings**.
>
> > [W2] Research data: No code is provided Experiment results are not included, e.g. as csv files
> >
>
> We appreciate your concern about the research data, but we would like to highlight that,
>
> - Appendix B contains **extensive experimental results**, mainly including comparisons of teacher models with varying accuracies and more numerical results, which we believe is **clear and sufficient enough** to support the key findings.
> - We also commit to releasing the complete code upon acceptance of the paper.
>
> > [W3] While interesting experiments are designed and phenomena are observed, little explanation is offered as to why these patterns are being observed. For example, in the context of Figure 7 it would be interesting to discuss the effects the different augmentation strategies have on the class probabilities, which might explain why some augmentation methods perform worse with strategy C than with strategy B. When an augmentation method actually changes the class probability of an image (such as random cropping), using strategy C to train the network does not reduce noise but instead feeds the network wrong soft labels.
> >
>
> In Figure 7, we agree with your opinion that for augmentation methods that significantly affect class probabilities (e.g., random cropping), STRATEGY C may provide incorrect soft targets. And we appreciate your suggestion to provide a deeper theoretical explanation for the observed patterns.
>
> However, we would like to emphasize that our current paper primarily focuses on **uncovering key factors influencing scaling laws** that were previously **unaddressed or insufficiently discussed**. While theoretical explanation is undoubtedly valuable, it is often the case in the deep learning community that **empirical observations precede and inspire theoretical studies**. For example:
>
> - In He et al.'s ResNet work [1], skip connections were empirically shown to mitigate gradient vanishing issues prior to detailed theoretical analysis.
> - Contrastive learning methods, such as SimCLR [2], were initially evaluated through empirical results before rigorous theoretical frameworks were developed.
> - The GPT-3 work [3], where scaling phenomena were identified and validated empirically, lacked a complete theoretical understanding at the time of publication.
>
> Thus, we believe our results similarly highlight a **highly non-trivial** open problem that **merits further exploration**. We identify this as an **important direction for future work**.
>
> [1] Deep Residual Learning for Image Recognition. CVPR 2016.
>
> [2] A Simple Framework for Contrastive Learning of Visual Representations. ICML 2020.
>
> [3] Language Models Are Few-Shot Learners. NeurIPS 2020.

---

> > ### Author Response · Authors · 2024-11-25
> >
> > > [Q1] Figure 1: How are the results obtained? For example, how were the number of epochs determined?
> > >
> >
> > Thank you for your valuable feedback. We have addressed this point in the revised version, specifically on line 715, page 14, where we clarify the number of training steps required for Figure 1. As for other hyper parameters:
> >
> > - batch size detailed in line 708
> > - learning rate detailed in line 740
> > - training step is set for 10000 steps
> >
> >     The training step keeps the same configuration as in Section 5, since:
> >
> >     - we aimed to ensure consistency across experiments for comparability
> >     - 10000 steps **is sufficient for the model to reach convergence on the small datasets**
> >
> > For other experiments, preliminary experiments indicated that:
> >
> > - training for 50 epochs was sufficient for the model to **reach convergence** in Figure 3a.
> > - training for 150 epochs was enough to **observe the difference between Strategy B and C.**
> >
> > Extending the training beyond this point **did not result in significant performance improvements** but substantially **increased computational cost**.
> >
> > > [Q2] Section 3.2: I do not fully understand the motivation behind the design of a unified loss function. It is said that CE is unable to exploit information from soft targets, but CE can be used with soft targets, so it’s not clear to me what is meant with that statement. Also, why is it not a valid evaluation strategy to evaluate strategies B and C by training the network exclusively using the soft labels with cross-entropy? Later it is said that KL divergence is used to leverage the information in the soft targets, but this seems to contradict what was said above, i.e. that CE cannot make use of soft labels, but minimizing CE and KL divergence is equivalent.
> > >
> >
> > Thank you for raising this concern. We appreciate the your comments that cross-entropy (CE) loss can technically be employed with soft labels. However, our motivation for designing a unified loss function stems from the following considerations:
> >
> > - **Prevalence of KL Loss in Knowledge Distillation:** In the context of knowledge distillation, the **predominant practice is to use Kullback-Leibler (KL) divergence** to incorporate the information from soft labels during model training. While CE loss can theoretically serve this purpose, KL divergence provides a more direct mechanism for measuring the similarity between the soft targets generated by the teacher model and the predictions of the student model.
> > - **Objective of the Proposed Loss Function:** Our primary aim in proposing the unified loss function is not merely to replace CE loss but to systematically **evaluate the three mapping strategies**. Conventional CE loss is **inherently coupled with the multiple-to-one mapping** (Strategy A, since it only mainly deal with one-hot target), making it challenging to decouple and assess the unique representational contributions of Strategy B and C. By separating the influence of soft targets on the backbone and classifier, the unified loss function provides a more transparent framework for analyzing these strategies.
> > - **Comparative Study Beyond Training Accuracy:** While evaluating Strategy B and Strategy C solely through CE loss with soft labels is a valid approach, it does not align with our research objective. Our focus is on understanding the impact of each mapping strategy on the **representational capacity of the backbone**. The unified loss function facilitates this analysis by isolating the effects of mapping strategies on backbone training, independent of the classifier's performance.
> >
> > We hope these clarifications address the concerns raised. The design of the unified loss function is intended to enhance our ability to study the effects of different mapping strategies systematically and is not merely a replacement for CE loss in standard settings.

---

> > > ### Author Response · Authors · 2024-11-25
> > >
> > > > [Q3] Figure 5: Which strategy was used to train the student?
> > > >
> > >
> > > As stated in Appendix B (line 768 on page 15), the student model in Figure 5a was trained using the **StdAug** augmentation strategy.
> > >
> > > > [Q4] On line 405 it says: “MixUp-trained teacher models achieve superior performance compared to standard augmentation”, but this is not supported by Figure 5a?
> > > >
> > >
> > > We acknowledge the typos in the description on line 405. The correct statement should be:
> > >
> > > - *For high-accuracy teacher models, applying MixUp augmentation to the teacher does not significantly benefit the student model’s training and may even perform worse than using a NoAug-trained teacher model.*
> > >
> > > This is also an **intriguing and novel** observation😊 and we have corrected this in the revised version.
> > >
> > > > [Q5] Why are the results of the experiments from the appendix not summarized in the main paper, i.e. whether they support the six key findings?
> > > >
> > >
> > > To clarify, the experiments in the appendix:
> > >
> > > - Primarily serve to supplement the main paper by **providing additional details**, such as those involving **more teacher models** and specific **numerical results.**
> > > - All figures and tables in the appendix are **explicitly referenced in the main body** and are **consistent with the six key findings** we proposed, such as in the **footnotes** on pages 7, 8, and 9.
> > >
> > > Therefore, we firmly believe that the experimental results in the appendix further strengthen the reliability and robustness of the findings presented in the main body of the paper.

---

> > > > ### Author Response · Authors · 2024-11-30
> > > > **A Kind Reminder for Reviewer apkJ**
> > > >
> > > > **Dear Reviewer apkJ,**
> > > >
> > > > Thank you for your detailed review and thoughtful feedback on our submission. Your comments have been instrumental in refining our work, and we have made significant revisions to address your concerns. Below, we summarize your key points and the corresponding updates in the revised version:
> > > >
> > > > 1. **Learning Rate and Error Quantification for Strategy A**
> > > >     - Your concern: Strategy A in Figure 3a might need a different learning rate, and uncertainty quantification (e.g., standard error) is missing.
> > > >     - Our response:
> > > >         - We conducted additional experiments with varying learning rates and batch sizes for Strategy A, as shown in Figure 12. These results confirm that tuning hyperparameters **does not alter the observed trends**.
> > > >         - Standard error values for repeated experiments are now included in Table 4 as a case study to demonstrate the robustness of our findings.
> > > > 2. **Research Data Availability**
> > > >     - Your concern: No code or raw experimental data is provided.
> > > >     - Our response:
> > > >         - We commit to releasing the complete code upon acceptance of the paper.
> > > >         - Extensive experimental results, including comparisons across teacher models with varying accuracies, are included in Appendix B. These results are ***DETAILED*** and ***SUFFICIENT*** to support the key findings.
> > > > 3. **Explanation of Observed Phenomena (e.g., Figure 7)**
> > > >     - Your concern: More theoretical discussion is needed, especially on the impact of augmentation strategies on class probabilities.
> > > >     - Our response:
> > > >         - We agree that for augmentation methods altering class probabilities (e.g., random cropping), Strategy C may introduce incorrect soft targets. While more rigorous theoretical analysis is ***BEYOND* the scope of this paper**, we recognize its importance and have identified this as a ***PROMISING*** direction for **future work**.
> > > > 4. **Details and Clarifications:**
> > > >     - Your concern:
> > > >         - Figure 1 lacks details.
> > > >         - Which strategy was used for training the student model in Figure 5?
> > > >         - The description of MixUp in line 405 contradicts Figure 5a.
> > > >         - Results in the appendix should be summarized in the main paper.
> > > >         - The motivation behind the unified loss function in Section 3.2 is unclear.
> > > >     - Our response:
> > > >         - Training steps for Figure 1 are clarified in line 715 (page 14), and other hyperparameters remain consistent across experiments.
> > > >         - As noted in Appendix B (line 768, page 15), the student model in Figure 5a was trained using the **StdAug** strategy.
> > > >         - We corrected the typo on line 405.
> > > >         - All appendix results supplement the main paper’s findings and are ***EXPLICITLY REFERENCED***, supporting the six key conclusions.
> > > >         - The unified loss function separates the backbone’s representational capacity from the classifier's influence, allowing a systematic evaluation of mapping strategies. We emphasized that while cross-entropy can handle soft labels, KL divergence ***BETTER ALIGNS*** with the objectives of knowledge distillation.
> > > >
> > > > We hope our clarifications and revisions adequately address your concerns. If the updates align with your expectations, we kindly request you to consider revising your score, as this would greatly support the progression of our work.
> > > >
> > > > Please let us know if you have additional questions or suggestions. We sincerely appreciate your thoughtful engagement and detailed review.
> > > >
> > > > **Best regards,**
> > > >
> > > > The Authors of Submission 10518

---

> > > > > ### Author Response · Authors · 2024-12-01
> > > > > **Eagerly Anticipating Reviewer apkJ’s Feedback**
> > > > >
> > > > > **Dear Reviewer apkJ,**
> > > > >
> > > > > We hope this message finds you well. As the discussion period draws to a close on December 2nd, we wanted to follow up to kindly ask if our responses have sufficiently addressed your feedback. If there are any remaining concerns or points for clarification, we would be glad to discuss them further.
> > > > >
> > > > > Thank you again for your time and effort in reviewing our submission!
> > > > >
> > > > > **Best regards,**
> > > > >
> > > > > The Authors of Submission 10518

---

### Official Review · Reviewer_wGQ7 · 2024-11-09

**Soundness:** 3
**Presentation:** 3
**Contribution:** 3
**Rating:** 6
**Confidence:** 3

**Summary:**

This paper investigates the often-overlooked role of targets in data-efficient learning, particularly in the context of neural scaling laws. Neural scaling laws indicate that achieving lower test errors typically requires exponentially more data and computational resources, leading to inefficiencies and sustainability challenges.

The authors observe that using soft targets on a smaller subset of data can outperform using one-hot targets on the full dataset under the same training budget. Motivated by this, they explore the impact of different sample-to-target mapping strategies on training efficiency. They categorize these strategies into three types:

STRATEGY A: Multiple augmented samples within the same class are mapped to a single one-hot target (conventional supervised learning).
STRATEGY B: Each augmented sample is mapped to a unique soft target generated by a teacher model (knowledge distillation).
STRATEGY C: Multiple augmented views of a single sample are mapped to the same soft target (proposed method to reduce noise in soft targets).

**Strengths:**

Comprehensive Analysis: The paper provides a thorough investigation of how different sample-to-target mappings and data augmentation strategies affect training efficiency, offering valuable insights.

Novel Perspective on Targets: By highlighting the often-neglected role of targets in dataset design, the paper contributes to a more holistic understanding of data-efficient learning.

Unified Loss Framework: The introduction of a unified loss function that separates the backbone training from the classifier training allows for a clearer evaluation of the representational capacity influenced by different strategies.

Practical Implications: The findings offer actionable guidance on selecting target types, teacher models, and augmentation strategies to enhance training efficiency, which can be beneficial for practitioners.

Extensive Experiments: The use of multiple datasets and varied experimental settings strengthens the validity of the conclusions drawn.

**Weaknesses:**

Theoretical Analysis: It would be ideal to provide a theoretical framework or intuition to explain the empirical observations, especially concerning why weaker teacher models can aid early learning and why STRATEGY C effectively reduces noise.

This addition would be a nice enhancement rather than any requirement, but I am not allowed to leave this section blank.🥸

**Questions:**

Applicability to Larger Datasets: Have you considered applying STRATEGY C to larger-scale datasets

============ Revise according to the Associate Program Chair's comments ================ $\searrow$

"Have you explored applying STRATEGY C to larger datasets like ImageNet? What computational or methodological challenges do you anticipate in scaling up this approach?"

Teacher Model Selection: How does the choice of teacher model architecture impact the student model's performance under STRATEGY B and STRATEGY C? $\searrow$

"Have you considered comparing the impact of different teacher model architectures (e.g., ResNet vs. Vision Transformer) on student performance under STRATEGY B and STRATEGY C?

---

> ### Author Response · Authors · 2024-11-25
>
> > [W1] Theoretical Analysis: It would be ideal to provide a theoretical framework or intuition to explain the empirical observations, especially concerning why weaker teacher models can aid early learning and why STRATEGY C effectively reduces noise.
> >
>
> Thank you for your insightful feedback and interest in the theoretical aspects of our work, and we would also like to emphasize that **the primary objective of this paper is to propose the novel Strategy C and systematically reveal its significance.** While theoretical explanation is undoubtedly valuable, it is common in the deep learning community for empirical observations to precede and inspire theoretical studies, for example:
>
> - Similar to the empirical findings in He's ResNet work in [1], where skip connections were observed to mitigate gradient vanishing issues prior to a detailed theoretical explanation.
> - Contrastive learning methods, such as [2], were initially evaluated based on empirical results without rigorous theoretical justification.
> - GPT-3 work[3], where the authors identified and empirically validated scaling phenomena without yet providing a complete theoretical understanding.
>
> Thus, we believe our results similarly highlight an important open problem that **merits further exploration**. However, we are willing to provide some **intuitive explanations** to address the two key questions raised:
>
> - **Why weaker teacher models can aid early learning?**
>     - Compared to a highly accurate teacher model, a weaker teacher model typically outputs **higher-entropy probability distributions** than a stronger teacher. This increases the KL divergence between the teacher and student models at the beginning of training, leading to larger gradients and **faster updates** of the student model parameters.
>     - However, as the teacher's accuracy decreases excessively (e.g., nearing uniform random predictions), the output target of the teacher model **approaches a uniform distribution**, reducing its informational content.
>
>     Consequently, **relatively weaker** teacher models strike a **balance**: they provide **sufficient signal to guide early learning while avoiding over-saturating** the student model with overly confident predictions.
>
> - **Why does STRATEGY C effectively reduce noise?**
>
>     As noted in line 344 of the paper, **STRATEGY C generates fewer distinct targets** during training compared to STRATEGY B. For example, on CIFAR-10 with 100 training epochs, the ratios of sample-to-target mappings are:
>
>     - STRATEGY A: 50,000×100:10=500,000:1,
>     - STRATEGY B: 50,000×100:50,000=1:1,
>     - STRATEGY C: 50,000×100:50,000=100:1.
>
>     By reducing the number of distinct targets, STRATEGY C effectively minimizes the uncertainty introduced by noisier (more) targets. Our proposed Strategic C **effectively addresses the short-term limitations** of Strategy A and the **long-term deficiencies** of Strategy B,  **striking a balance between short-term acceleration and long-term convergence.**
>
>
> In a word, we believe this work is **a step toward understanding how target mappings impact training dynamics** across domains.
>
> [1] Deep Residual Learning for Image Recognition. CVPR 2016.
>
> [2] A Simple Framework for Contrastive Learning of Visual Representations. ICML 2020.
>
> [3] Language Models Are Few-Shot Learners. NeurIPS 2020.

---

> ### Author Response · Authors · 2024-11-25
>
> > [Q1] "Have you explored applying STRATEGY C to larger datasets like ImageNet? What computational or methodological challenges do you anticipate in scaling up this approach?" Teacher Model Selection: How does the choice of teacher model architecture impact the student model's performance under STRATEGY B and STRATEGY C? $\searrow$ "Have you considered comparing the impact of different teacher model architectures (e.g., ResNet vs. Vision Transformer) on student performance under STRATEGY B and STRATEGY C?
> >
>
> **Yes,** we have explored applying STRATEGY B and STRATEGY C to larger datasets on **ImageNet,** and **we also use ResNet50 and ViT as the backbone for model training**, as shown in Figure 15 and Table 6, 7 for numerical results, Under different backbone settings, we also find that:
>
> - Strategy A shows long-term advantages, while Strategy B exhibits short-term benefits.
> - Our proposed Strategy C **effectively addresses the short-term limitations of Strategy A and the long-term deficiencies of Strategy B**.
> - Moreover, the advantages of Strategy C become **increasingly prominent when applied to weaker teacher models**.
>
> Regarding the **computational  challenges**:
>
> - Within the 250k training steps we tested, we have already found the **performance improvements of STRATEGY C over Strategy B** under different teachers.
> - As discussed in [R1], the **efficacy of STRATEGY C** is highly dependent **on the number of training steps**, since STRATEGY C establishes a mapping between samples and targets proportional to the number of training epochs.
>
>     Since we can find in the table that the Gain Strategy C over Strategy B consistently increases as the training process, we believe that the noise-reduction effect will **become more significant with longer training steps**.
>
>
> Regarding the impact of **different model architectures**:
>
> - The numerical results show that, due to the larger number of parameters in ViT compared to ResNet50, using **ViT as the backbone** to train the student model under Strategy C **leads to more significant performance improvements** within the same computational budget (i.e., the same number of training steps).
>
> Regarding the **scalability:**
>
> - In Section 6, we briefly highlight the **applicability** of our findings
>     - We can extend our findings to larger-scale datasets like text-based tasks. This paper uses image classification **as a case study** to provide a concrete and focused demonstration of the proposed methods, with broader implications left for future exploration.
>     - We would also like to emphasize that this work is **the first to systematically emphasize that a weak teacher model can significantly contribute to the training of student models** in the context of knowledge distillation.
>
>     We believe this further supports the validity of our findings and their applicability under more settings.

---

> > ### Author Response · Authors · 2024-11-30
> > **A Kind Reminder for Reviewer wGQ7**
> >
> > **Dear Reviewer wGQ7,**
> >
> > Thank you for your thoughtful review and positive assessment of our submission. We greatly appreciate your constructive feedback and have addressed your comments in the revised version. Below, we summarize the key points and our explanation:
> >
> > 1. **Theoretical Analysis**
> >     - Your concern: Additional theoretical explanations for why weaker teacher models aid early learning and how STRATEGY C reduces noise would be beneficial.
> >     - Our response:
> >         - **Weaker teacher models**: We have provided intuitive above that weaker teacher models output **higher-entropy distributions**, yielding larger gradients that accelerate early training. However, excessively weak teachers (e.g., random predictions) **reduce informational content**.
> >         - **Noise reduction in STRATEGY C**: We emphasized how STRATEGY C generates **fewer distinct targets** than STRATEGY B, thereby reducing noise while retaining sufficient target variety to improve convergence.
> > 2. **Applicability to Larger Datasets & Teacher Model Architecture**
> >     - Your concern: Can we applied STRATEGY C to larger datasets like ImageNet, and how do different teacher model architectures (e.g., ResNet vs. ViT) influence performance under STRATEGIES B and C?
> >     - Our response:
> >         - We extended our experiments to ImageNet with ResNet50 and ViT backbones, as shown in Figure 15 and Tables 6 and 7. These results confirm STRATEGY C’s effectiveness in improving training efficiency and accuracy at larger scales.
> >
> > We hope our revisions and responses adequately address your feedback, and we also ***further summarize and emphasize the main contributions of our paper in the General Response***.
> >
> > If the updates align with your expectations, we kindly request that you consider confirming or revising your score to reflect the improvements, as this would greatly support the progression of our work. We remain open to further questions or suggestions and sincerely appreciate your time and effort in reviewing our submission.
> >
> > **Best regards,**
> >
> > The Authors of Submission 10518

---

> > > ### Author Response · Authors · 2024-12-01
> > > **Eagerly Anticipating Reviewer wGQ7’s Feedback**
> > >
> > > **Dear Reviewer wGQ7,**
> > >
> > > Thank you for your insightful feedback and for your valuable contribution to the review process. As the discussion period is nearing its conclusion (December 2nd), we kindly ask if our responses have addressed your concerns. If you have any further questions or comments, we would be glad to continue the discussion.
> > >
> > > Thank you once again for your time and expertise!
> > >
> > > **Best regards,**
> > >
> > > The Authors of Submission 10518

---

### Author Response · Authors · 2024-11-25
**General Response**

We sincerely thank all reviewers for their constructive feedback and thoughtful evaluations. We are encouraged by their recognition of various strengths in our work and have taken their insights into account to enhance the **clarity and impact** of our paper. Below, we summarize the key points acknowledged by the reviewers:

1. **Novel Perspective on Neural Scaling Laws and Training Efficiency**:
    - We highlight the often-overlooked role of targets in dataset design and how **redefining sample-to-target mappings can improve data efficiency**. This introduces a **meaningful and new training framework** that, to the best of our knowledge, has not been proposed before. This perspective on targets has been acknowledged as **novel and insightful** by Reviewer wGQ7 and aligns with the relevance of neural scaling laws noted by Reviewers pm2n and trqj.
2. **Comprehensive Evaluation of Target-Mapping Strategies**:
    - We systematically compare three strategies (A, B, C), revealing:
        - Strategy A exhibits **long-term advantages** but struggles in early training.
        - Strategy B **accelerates early convergence** but is constrained by the teacher model's capacity.
        - Strategy C **effectively addresses the short-term limitations** of Strategy A and the **long-term deficiencies** of Strategy B.
    - These findings are validated on **diverse datasets** (e.g., CIFAR-10/100, ImageNet) and **different architectures** (ResNet50, ViT), as noted by Reviewer wGQ7, who appreciated the robustness of our experiments.
3.  **Broader Applications And Strong Practical Implications**:
    - The proposed strategies, particularly Strategy C, offer a **viable approach to mitigating inefficiencies** in traditional training paradigms.
    - To the best of our knowledge, systematic studies focusing on the benefits of weaker teacher models are scarce in knowledge distillation, and we are the first to reveal that **weaker teacher models can significantly enhance student model performance,** an insight validated by Reviewer apkJ’s emphasis on the impact of teacher quality.
    - Our findings provide ***ACTIONABLE INSIGHTS*** for deep learning community on how to select ***TARGET TYPES, TEACHER MODELS***, and ***AUGMENTATION STRATEGIES*** to enhance training efficiency, as highlighted by Reviewers wGQ7 and apkJ.
4. **Future Impact and Directions**:
    - By introducing a novel focus on target-mapping strategies, our work opens up a **promising** direction for further research, including **theoretical studies** and **applications to other domains**.
    - We believe this work is **a step** toward **understanding how target mappings influence training dynamics** and addressing the inefficiencies highlighted by neural scaling laws, a concern emphasized by Reviewers pm2n and trqj.

To address the reviewers' concerns, we have made significant revisions, including:

- **Scaling to Larger Datasets and Architectures**: We extended our experiments to larger datasets, such as **ImageNet**, and employed additional backbones **(ResNet50 and ViT)**. These results are included in Figure 15 and Table 6, 7.

While theoretical analyses are valuable, we emphasize that our primary objective is to ***SYSTEMATICALLY*** analyse the ***CRITICAL*** and often ***OVERLOOKED*** role of ***TARGETS AND THEIR MAPPINGS*** in a dataset, thus propose and validate a ***MEANINGFUL*** and ***NOVEL*** training paradigm that benefits **efficient learning**, with theoretical exploration left as a **promising direction for future work**.

---

### Meta-Review · Area_Chair_Dyyv · 2024-12-21

**Metareview:**

The paper studies the effectiveness of training with soft labels. Authors position their paper in the context of neural scaling laws, where diminishing returns of various kinds have been established. The authors conducted an interesting set of experiments and generated valuable insights from those. However, the experiments, including the additional ones with other backbones (ResNet50 and ViT), do not show that strategy C, the one advocated by the authors, exhibits the best of both worlds as Strategy A (the conventional strategy) remains superior when more training budget is available. Hence, while the observation that soft labels can boost the training speed initially, one could argue that they negatively impact the final performance of the model. The authors do not propose an explanation or a more theoretical justification of the assumed benefits of Strategy C. In addition, questions remain how Strategy C is actually implemented (and the impact of the actual implementation): for instance, depending on the augmentation family (eg, MixUp), it is not clear how a single soft label would be produced for all augmented variants of a data point. Overall, the paper is well written and the direction interesting, but the paper is lacking evidence to support the main claims.

**Additional Comments On Reviewer Discussion:**

Reviewers with higher confidence remained critical post rebuttal. One reason is that across every experiment when a large amount of data and computation are available (i.e., in the regime relevant for neural scaling laws) standard training with 1-hot labels remained the best approach. Another aspect that remained a concern is that when moat augmentation methods change the class probability of an the data with strategy C. This means that the network is actually fed wrong (soft) labels, which is probably explaining the overall superiority of Strategy A in the end.

---

### Decision · Program_Chairs · 2025-01-22

Reject